# Stability Matters: Combating Parameter Shifts in Low-Rank Adaptation for Continual Learning

## Abstract

Continual Learning (CL) has increasingly embraced Parameter-Efficient Fine-Tuning (PEFT) methods, particularly Low-Rank Adaptation (LoRA), to balance task adaptability with parameter efficiency. Existing LoRA-based approaches resort to low-rank matrices to inherently capture task-specific parameter shifts, whereas meantime mitigate interference between tasks through architectural design (e.g., Mixture-of-Experts) or optimization constraint (e.g., orthogonality). However, they largely overlook how these shifts evolve across tasks, i.e., the internal dynamics of parameter space, which is a crucial yet underexplored factor in model forgetting. In this work, our analysis reveals a key insight that abrupt performance drops often coincide with drastic changes in the distribution of learned parameter shifts. Motivated by this, we propose a simple yet effective *Parameter Stability Loss* that regularizes both the sign and magnitude of parameter updates to mitigate forgetting. Beyond training-time regularization, we also introduce a post-training model merging step that bridges earlier directions with the current one and further combats the inevitable drift toward new tasks. Our method **Parameter Stable LoRA (PS-LoRA)** achieves state-of-the-art results on multiple continual learning benchmarks, with performance improvements of up to 3%, and can be integrated with existing approaches.

## 1 Introduction

Continual Learning (CL) (Parisi et al., 2019; Wang et al., 2024a; Wu et al., 2024) has emerged as a crucial paradigm in natural language processing (NLP), where models are expected to learn from a sequence of tasks without forgetting previously acquired knowledge. As NLP systems are increasingly deployed in dynamic, real-world environments such as dialogue systems (Li et al., 2022), personalized assistants (Yu et al., 2024a), and evolving domain applications (Chuang et al.), they must adapt to new information over time while maintaining performance on earlier tasks. While large pre-trained language models have shown remarkable success on static benchmarks (Brown et al., 2020; Devlin et al., 2019; Raffel et al., 2020; Touvron et al., 2023), how to mitigate the notorious *catastrophic forgetting* (McCloskey & Cohen, 1989) problem (i.e., losing knowledge learned from earlier tasks) when trained sequentially on multiple tasks remains a daunting challenge.

Unlike traditional CL methods (Zenke et al., 2017; Kirkpatrick et al., 2017; Li & Hoiem, 2017) that train models from scratch, recent approaches emphasize efficiently leveraging pre-trained models to better mitigate forgetting. Specifically, state-of-the-art CL approaches increasingly adopt and customize the Low-Rank Adaptation (LoRA)(Hu et al., 2022) strategy for sequential training, aiming to reduce parameter interference and mitigate forgetting. For instance, AM-LoRA (Liu et al., 2024), MoCL (Wang et al., 2024b) and MoeLoRA(Yu et al., 2024b) follow a Mixture-of-Experts (MoE) (Jacobs et al., 1991) paradigm, selecting task-specific low-rank matrices at inference time to enhance prediction accuracy from an architectural perspective. In contrast, InfLoRA (Liang & Li, 2024) and O-LoRA (Wang et al., 2023b) impose orthogonality constraints on the low-rank matrices to address forgetting from an optimization perspective. While both approaches are effective, they differ in managing parameter updates. MoE-based approaches aggregate task-specific LoRA weights via attention mechanisms, whereas orthogonality-based methods regulate LoRA parameter updates by constraining gradient update directions.

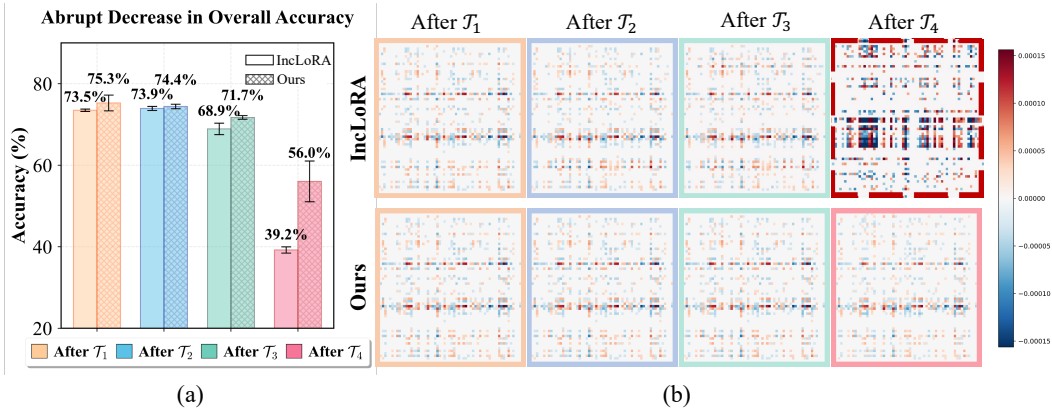

Figure 1: Comparison between incremental LoRA training and our method. (a) shows the average accuracy on all seen tasks after training on the $i$-th task $\mathcal{T}_i$. (b) visualizes the parameter shift distributions at each training stage for a randomly selected representative layer of the pre-trained model. More detailed results about different task orders and parameter shifts please see Appendix B.

However, neither method directly examines how parameter shifts evolve across tasks (*i.e.*, the internal dynamics of parameter space), which is a crucial yet underexplored factor in model forgetting. To shed light on this underexplored aspect, we begin with a CL task in the NLP domain. During incremental LoRA finetuning, we observe that certain tasks cause the model to abruptly forget previously learned knowledge, leading to a sudden drop in the overall average accuracy (e.g. after $\mathcal{T}_4$ in Fig. 1 (a)). Different from prior LoRA-based CL approaches (Liu et al., 2024; Wang et al., 2024b; Wu et al., 2025), we directly examine the parameter shift across tasks during training. Interestingly, we find that the abrupt performance drop is consistently accompanied by rapid shifts in the distribution of model parameters in LoRA's subspace. As shown in Fig. 1 (b), the parameter shift pattern after training on task $\mathcal{T}_4$ in the Incremental LoRA method exhibits a clear deviation from earlier tasks, coinciding with the largest performance drop shown in Fig. 1 (a). We further verify this phenomenon across different task orders, consistently observing the same patterns. Detailed analyses are provided in Appendix B.

Building on the observed correlation between forgetting and parameter shift in Fig. 1, we further decouple the parameter shift into two components: magnitude and parameter-wise sign direction (*i.e.*, positive or negative), and apply targeted regularization to each component. We find that jointly constraining both aspects effectively mitigates forgetting, particularly by reducing abrupt performance drops during CL. In summary, our main contributions are as follows:

- We observe abrupt and severe forgetting during sequential training, closely tied to large shifts in parameter space. Analyzing this in terms of magnitude and sign, we find that large updates with opposite signs can reverse parameter directions, resulting in sharp performance drops.

- We propose a simple yet effective Parameter Stability Loss, which not only prevents reverse parameter updates during LoRA training and mitigates forgetting, but also facilitates synergy with state-of-the-art model merging strategies during inference to further boost CL performance.

- We conduct evaluations on diverse CL NLP and CV benchmarks with varying tasks, lengths and orders, and our method achieves up to a 3% improvement over existing leading approaches.

## 2 RELATED WORK

**Continual Learning** aims to adapt models to new sequential tasks while maintaining the previously acquired knowledge. Existing methods can be broadly categorized into regularization-based (Dhar et al., 2019; Li & Hoiem, 2017; Kirkpatrick et al., 2017), optimization-based (Farajtabar et al., 2020) and architecture-based methods (Wu et al., 2025; Liu et al., 2024; Wang et al., 2024b; Razdaibiedina et al., 2023; Wang et al., 2022b; Qiao & Mahdavi, 2024).

- *Regularization-based methods* typically identify important weights and introduce penalty terms to protect them so as to mitigate forgetting. For example, LwF (Li & Hoiem, 2017) preserves previously learned knowledge by constraining the outputs of the model on old tasks while fine-tuning it on new ones. In contrast, methods such as EWC (Kirkpatrick et al., 2017), IS (Zenke

et al., 2017), KFLA (Ritter et al., 2018), and VR-MCL (Wu et al., 2024) estimate the Hessian matrix through different techniques to identify and protect the important weights.

- *Optimization-based methodes* aim to improve knowledge retention by projecting update gradients or constraining the weight update space. For instance, O-LoRA (Wang et al., 2023b) and InfLoRA (Liang & Li, 2024) both impose orthogonal constraints on the learnable low-rank matrix to minimize interference, while MIGU (Du et al., 2024) restricts updates to parameters with the largest gradient magnitudes.
- *Architecture-based methods* design specific module architectures to help model learning and alleviate catastrophic forgetting. For instance, MoE mechanism allocates or selects task-specific parameter subsets and route inputs accordingly. In the context of PEFT, prompt-based methods L2P and DualPrompt (Wang et al., 2022b;a) maintain a bank of prompts chosen per task, while LoRA-based methods such as MoCL (Wang et al., 2024b) and AM-LoRA (Liu et al., 2024) dynamically combine multiple LoRA modules to reduce interference via task-aware routing. These mechanisms reduce inter-task interference across different parameter locations.

Our proposed method extends the research line of optimization-based approaches. Unlike prior work that uses orthogonality and MoE approach addressing different position or direction parameter collision, we conduct a parameter-wise analysis during training, identifying a key issue in incremental learning that large shifts in parameter distributions lead to forgetting and address it effectively.

**Model Merging** has become a *de-facto* practice in multi-task learning with large foundation models (Raffel et al., 2020; Touvron et al., 2023; Devlin et al., 2019). Different from traditional multi-task learning, which jointly updates the full model by weighting gradients from multiple tasks, model merging focuses on extracting task-specific parameter shift, such as LoRA, and combining these shifted parameters while keeping the shared backbone frozen. These approaches enable efficient knowledge transfer by applying various merging strategies to external memory components trained independently on different tasks. For example, Task Arithmetic (Ilharco et al., 2023) merges task vectors obtained through task-specific fine-tuning using direct interpolation. Ties-Merging (Yadav et al., 2023) and Fisher-merge (Matena & Raffel, 2022) demonstrate that sparsity and parameter sign play a critical role in the effectiveness of merging. Furthermore, MagMax (Marczak et al., 2024) highlights the importance of parameter magnitudes in model merging. However, most model merging approaches primarily focus on merging multi-task learning task vectors, with limited attention to achieving the adaptation-retention trade-off in CL scenarios.

## 3 METHOD

### 3.1 PRELIMINARIES

**Continual Learning Setup.** Suppose there are $N$ sequential tasks $\{\mathcal{T}_1, \mathcal{T}_2, \ldots, \mathcal{T}_N\}$, where each task $\mathcal{T}_t$ is associated with a training dataset $\mathcal{D}_t = \{(\mathbf{x}_i^{(t)}, y_i^{(t)})\}_{i=1}^{|\mathcal{D}_t|}$ containing $|\mathcal{D}_t|$ examples. Let $f_\theta(\cdot)$ denote the predictive model parametrized by $\theta$. Since samples from historical tasks are inaccessible, the loss function for CL when training on current task $\mathcal{T}_t$ is given by:

$$\mathcal{L}_f = \sum_{(\mathbf{x},y)\in\mathcal{D}_t} -\log f_\theta(y \mid x). \tag{1}$$

**Low-Rank Adaptation (LoRA).** Given a pre-trained fixed weight matrix $\mathbf{W}_0 \in \mathbb{R}^{d\times k}$, LoRA (Hu et al., 2022) constrains the weight update $\Delta\mathbf{W}$ by representing it as a product of two low-rank matrices, enabling parameter efficient fine-tuning:

$$\mathbf{W} = \mathbf{W}_0 + \Delta\mathbf{W} = \mathbf{W}_0 + \mathbf{A}\mathbf{B}, \tag{2}$$

where $\mathbf{A} \in \mathbb{R}^{d\times r}, \mathbf{B} \in \mathbb{R}^{r\times k}$ are trainable parameters, and the rank $r \ll \min(d, k)$. During inference, the parameters $\Delta\mathbf{W}$ can be incorporated into $\mathbf{W}_0$ without introducing any extra computation cost.

For each task $\mathcal{T}_i$, fine-tuning yields a pair of corresponding low-rank matrices $\mathbf{A}_i\mathbf{B}_i$, which is qualified to be a task vector in model merging methods (Yadav et al., 2023; Matena & Raffel, 2022), capturing task-specific parameter shift and informing merging strategies to enhance overall performance.

### 3.2 MOTIVATION: LARGE-SCALE PARAMETER SHIFT

Rather than proposing an alternative to prior LoRA-based continual learning methods (Wang et al., 2023b; Liu et al., 2024; Wang et al., 2024b), which employ orthogonality constraints or MoE strategies

to mitigate forgetting, we complement these efforts by taking a parameter-wise perspective to examine the underlying training dynamics. Our analysis reveals that large shifts in parameter distributions, particularly excessive updates with opposite signs, are strongly associated with severe forgetting.

**Significant performance drop in CL is often aligned with large parameter distributional shifts.** As shown in Fig. 1 (a), we plot the training accuracy histogram of IncLoRA (defined in Eqn. (3)) over sequential tasks and observe a notable drop in average accuracy after learning task $\mathcal{T}_4$. This decline is a common phenomenon in CL (Caccia et al., 2022), with more examples provided in Appendix B. This huge decline aligns with a substantial shift in the distribution of LoRA parameters, as illustrated in Fig. 1 (b). Here, the visualized parameter distributions correspond to the cumulative parameter shift $\sum_{i=1}^{t} \Delta\mathbf{W}_i$ after task $\mathcal{T}_t$, reflecting the progressive evolution of LoRA updates relative to the frozen pre-trained model.

$$\mathbf{W} = \mathbf{W}_0 + \sum_{i=1}^{t} \Delta\mathbf{W}_i = \mathbf{W}_0 + \sum_{i=1}^{t-1} \mathbf{A}_i\mathbf{B}_i + \mathbf{A}_t\mathbf{B}_t. \tag{3}$$

We further investigate how parameter dynamics relate to the above large forgetting phenomenon. Building upon the analysis in Fig. 1 (b), we conduct a more fine-grained investigation, with the results presented in Fig. 2. We decouple the parameter shift $\Delta\mathbf{W}$ into two components: the previously learned $\Delta\mathbf{W}_i = \mathbf{A}_i\mathbf{B}_i$ and the newly learned $\Delta\mathbf{W}_t = \mathbf{A}_t\mathbf{B}_t$. Except for $\Delta\mathbf{W}_1$, which corresponds to the initial update and naturally exhibits a relatively large change, the LoRA parameters learned for tasks $\mathcal{T}_2$ and $\mathcal{T}_3$ (i.e., $\Delta\mathbf{W}_2$ and $\Delta\mathbf{W}_3$) show only minor shifts in parameter values. However, for $\mathcal{T}_4$, which leads to a sharp performance drop, we observe a substantial shift in the newly learned parameters $\Delta\mathbf{W}_4$. This indicates a strong correlation between significant parameter shifts and forgetting.

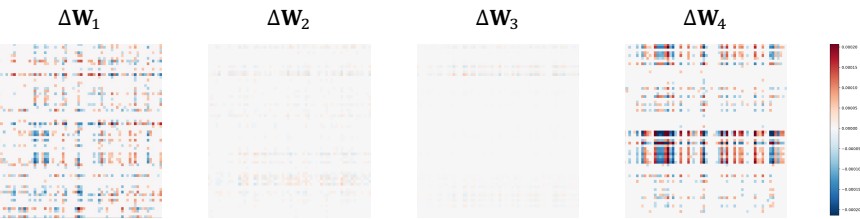

Figure 2: Detailed analysis of parameter shift in IncLoRA illustrated in Fig. 1(b), where $\Delta\mathbf{W}_i$ denotes the $i$-th specific learned $\mathbf{A}_i\mathbf{B}_i$ reflecting the parameter shift introduced by $\mathcal{T}_i$.

**Large updates with opposite signs flip the direction of parameters, causing a sharp performance decline**. To better understand how parameter changes affect performance, we perform a decomposition analysis on the update matrix $\Delta\mathbf{W}_t$ at task $\mathcal{T}_t$, separately examining the effects of sign and magnitude. Specifically, we select the bottom-$k\%$ parameters from $\Delta\mathbf{W}_t$. Then, we divide these parameters based on their sign consistency with the accumulated updates from previous tasks, i.e., $\sum_{i=1}^{t-1} \Delta\mathbf{W}_i$, yielding two subsets: $\Delta\mathbf{W}_t^{\text{sa}}$ and $\Delta\mathbf{W}_t^{\text{op}}$. We evaluate the performance using weight: $\mathbf{W} = \mathbf{W}_0 + \sum_{i=1}^{t-1} \Delta\mathbf{W}_i + \Delta\mathbf{W}_t^{\star}$ where $\star \in \{\text{same, opposite, both}\}$. The corresponding performance is shown in Fig. 3. It is evident that retaining only the same-sign parameters preserves high performance, while incorporating large opposite-sign

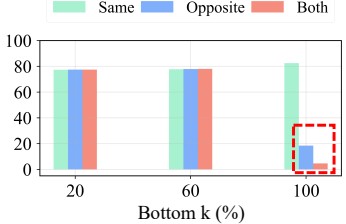

Figure 3: Evaluation results of different update subsets selected from the bottom-$k\%$ parameters of $\Delta\mathbf{W}_t$, analyzing the effects of sign consistency (same vs. opposite) and update magnitude on performance.

updates, as highlighted by the red box, leads to a substantial performance drop. However, manually removing parameters with conflicting signs after each task yields performance gains under a small number of tasks, it fails to prevent catastrophic forgetting as the task count grows. More details of this experiments are in Appendix D.1.

These findings motivate us to constrain model updates and prevent sign-flipping behaviors use a regularization-based method, thereby alleviating forgetting.

## 3.3 THE PROPOSED PS-LORA ALGORITHM

**Parameter Stability Loss**. Based on the observation in Fig. 3 that large sign-flipping parameter updates highly correlate with severe forgetting, we propose a parameter stability loss $\mathcal{L}_s$ to constrain such disruptive changes. As defined in Eqn. (4), the designed $\mathcal{L}_s$ consists of two items: (a) a *magnitude constraint term*, which applies L2 norm to the newly learned LoRA parameters $\mathbf{A}_t\mathbf{B}_t$ to suppress excessive updates; (b) a *sign alignment term*, which leverages the product $\tanh(\alpha \cdot (\mathbf{A}_t\mathbf{B}_t)) \cdot \tanh(\alpha \cdot (\sum_{i=1}^{t-1} \mathbf{A}_i\mathbf{B}_i))$ to encourage alignment in direction between new and previous updates, where $\alpha$ denotes the temperature parameter. When $\mathbf{A}_t\mathbf{B}_t$ and $\sum_{i=1}^{t-1} \mathbf{A}_i\mathbf{B}_i$ exhibit consistent element-wise signs, the resulting product tends toward 1, thus minimizing the associated loss.

$$\mathcal{L}_s = \underbrace{\|\mathbf{A}_t\mathbf{B}_t\|_2^2}_{(a)} \cdot \underbrace{\left(1 - \tanh\left(\alpha \cdot (\mathbf{A}_t\mathbf{B}_t)\right) \cdot \tanh(\alpha \cdot (\textstyle\sum_{i=1}^{t-1} \mathbf{A}_i\mathbf{B}_i))\right)}_{(b)}. \tag{4}$$

During sequential training, we simply combine the proposed $\mathcal{L}_s$ with the fine-tuning loss $\mathcal{L}_f$ shown in Eqn. (1). The overall training loss function $\mathcal{L}$ is given as $\mathcal{L} = \mathcal{L}_f + \lambda\mathcal{L}_s$, where $\lambda$ refers to a hyper-parameter. As shown in Table 6, adding $\mathcal{L}_s$ effectively mitigates large updates with opposite signs, thereby alleviating forgetting and improving the average accuracy.

According to the analysis in Sec. 3.2, we introduce the PS-LoRA algorithm, which combines a Parameter Stability loss to guide the training of LoRAs towards minimal parameter shifts. The resulting LoRAs are thus well tailored for a post-training merging strategy, further improving CL. The following paragraphs detail each component, and the overall procedure is listed in Algorithm 1.

As shown in Fig. 4, we visualize the angle between the LoRA directions of the most recent task and those of the initial task, where we quantify this similarity using the Frobenius inner product $\text{sim}(\mathbf{A}, \mathbf{B}) = \frac{\langle\mathbf{A},\mathbf{B}\rangle_F}{\|\mathbf{A}\|_F \|\mathbf{B}\|_F}$, where $\langle\mathbf{A}, \mathbf{B}\rangle_F = \text{Tr}(\mathbf{A}^\top\mathbf{B})$. The results confirm that our parameter stability loss significantly enhances stability, keeping LoRA updates closer to earlier directions. Meantime, we can still observe an inevitable shift in LoRA directions (highlighted in the red region), where the updates drift toward newer tasks after continual training. This naturally raises the question: **can we take a step further to bridge such shift?** Motivated by the superior performance of model merging in balancing between multiple tasks, we introduce a post-training model merging stage. This stage consolidates prior LoRA updates by realigning them toward an intermediate direction, thereby better retaining knowledge from earlier tasks.

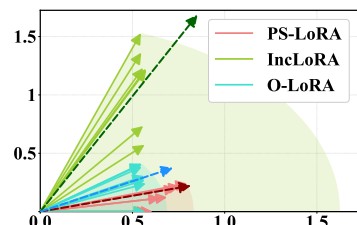

Figure 4: Distributions of different LoRAs. Vectors represent the LoRA directions; the angle between each vector and the axis indicates its deviation from the *earliest* task. *Vectors in dotted lines denote merged LoRAs.*

**Merging Strategies**. Building on prior model merging strategies and findings (Yadav et al., 2023; Marczak et al., 2024) that emphasize the importance of large-magnitude parameters in resolving merging conflict, we merge multiple LoRA weights $\Delta\mathbf{W}_i$ ($i \in 1, 2, \ldots, t$) obtained from sequential training and prioritize preserving parameters with higher magnitudes. As a post-training step, this merging strategy complements the training-time parameter stability loss, as described below,

$$\mathbf{W} = \mathbf{W}_0 + \Delta\mathbf{W}_{[1:t]} = \mathbf{W}_0 + \mathcal{M}(\Delta\mathbf{W}_{[1:t-1]}, \Delta\mathbf{W}_t),$$

$$[\mathcal{M}(\Delta\mathbf{W}_1, \Delta\mathbf{W}_2)]_{i,j} = \begin{cases} [\Delta\mathbf{W}_2]_{i,j}, & \text{if } |\,[\Delta\mathbf{W}_2]_{i,j}\,| \geq |\,[\Delta\mathbf{W}_1]_{i,j}\,| \\ [\Delta\mathbf{W}_1]_{i,j}, & \text{otherwise} \end{cases} \quad \text{for all } i, j, \tag{5}$$

where $\mathcal{M}(\cdot, \cdot)$ denotes an element-wise merging operation that selects, for each position $(i, j)$, the value with the larger absolute magnitude between two weight update matrices. And the notation $\Delta\mathbf{W}_{[1:t]}$ means the merged model LoRA matrices accumulated from all $t$ tasks.

**Remark**. The merging process only requires storing the current LoRA matrices and the previously merged version, making it efficient in both computation and memory. During inference, the merged weights can be directly integrated into the base model without introducing additional overhead. We defer a detailed analysis to Appendix D.8.

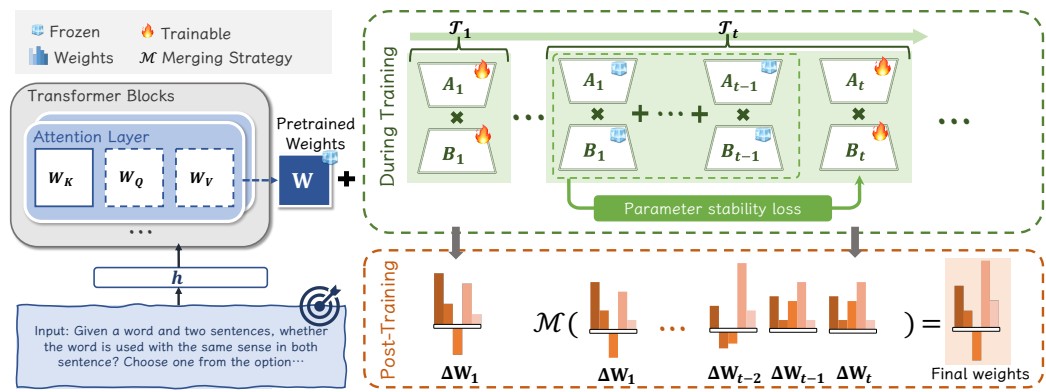

Figure 5: Overview of the proposed PS-LoRA. During training, Parameter Stability Loss is applied to the new LoRA to prevent sign-flip updates. After training, all LoRAs are merged by selecting the weights with the largest absolute magnitude and then added to the pre-trained model for inference.

---

**Algorithm 1:** The Proposed PS-LoRA Algorithm

---

**Input:** Pretrained weights $\mathbf{W}_0$, training datasets $\{\mathcal{D}_1, \ldots, \mathcal{D}_N\}$, hyper-parameters $r, \lambda$.
**Output:** Merged LoRA update $\Delta \mathbf{W}$
**for** $t = 1$ **to** $N$ **do**
    Initialize $\mathbf{A}_t \in \mathbb{R}^{d \times r}$, $\mathbf{B}_t \in \mathbb{R}^{r \times k}$;
    **for** *each minibatch* $(x, y) \in \mathcal{D}_t$ **do**
        Forward pass: $h = \mathbf{W}_0 x + \Delta \mathbf{W}_{[1:t-1]} x + \Delta \mathbf{W}_t x$;
        Compute the total loss $\mathcal{L} = \mathcal{L}_f + \lambda \mathcal{L}_s$ as defined in Eqn. (1) and Eqn. (4);
        Update $\mathbf{A}_t, \mathbf{B}_t$ by minimizing $\mathcal{L}$;
Merge all trained LoRAs (*i.e.*, $\Delta \mathbf{W}_i$, $i = 1, 2, .., t$) based on Eqn. (5) ;

---

### 3.4 THEORETICAL ANALYSIS

To provide theoretical justification for our method, we consider the simplest case of two sequential training datasets $\mathcal{D}_A$ and $\mathcal{D}_B$ . Let $\mathcal{L}$ denote the loss function, $\theta$ the model parameters, and $\theta_i^*$ the parameters after training on task $\mathcal{D}_i$. Assuming that $\mathcal{L}$ is twice differentiable and smooth near $\theta_A^*$, that the model has approximately converged to $\mathcal{D}_A$ (i.e., $\nabla \mathcal{L}_A(\theta_A^*) \approx 0$), and that parameter updates are small, the loss increase on $\mathcal{D}_A$ can be approximated by a second-order Taylor expansion:

$$\Delta \mathcal{L}_A \approx \tfrac{1}{2}(\theta - \theta_A^*)^\top \mathbf{H}_A (\theta - \theta_A^*), \tag{6}$$

where $\mathbf{H}_A$ is the Hessian of $\mathcal{L}_A$ at $\theta_A^*$. Since $\mathbf{H}_A$ is positive semi-definite near a local minimum, the increase admits the bound $\Delta \mathcal{L}_A \leq \tfrac{1}{2}\lambda_{\max}\|\theta - \theta_A^*\|^2$, with $\lambda_{\max}$ denoting the largest eigenvalue of $\mathbf{H}_A$. This result shows that forgetting depends on both the magnitude of parameter updates and the curvature of the loss surface. Our proposed PS-Loss is designed to mitigate this effect by explicitly constraining $\|\theta - \theta_A^*\|^2$ from two complementary perspectives: (i) a *sign constraint*, which prevents disruptive flips in parameter direction that can lead to functional interference with previous tasks, and (ii) a *magnitude constraint*, which limits the scale of parameter updates and thus reduces the risk of loss increase in high-curvature directions. Together, these constraints provide a principled mechanism to alleviate forgetting, consistent with the theoretical bound above.

## 4 EXPERIMENTS

**Benchmarks**. Following O-LoRA (Wang et al., 2023b), SD-LoRA(Wu et al., 2025) and Tree-LoRA (Qian et al., 2025), we evaluate on three widely used CL benchmarks across NLP and CV modalities: **Standard & Long, TRACE, and ViT Benchmark**. The **Standard & Long benchmark**, built from GLUE (Wang et al., 2018), SuperGLUE (Wang et al., 2019), and IMDB (Maas et al., 2011), provides task sequences of length 4 and 15 to assess short- and long-horizon NLP performance. The **TRACE** benchmark includes 8 sub-datasets covering multilingual tasks, code generation, and mathematical reasoning. For **ViT Benchmark**, we adopt Split ImageNet-R with varying task lengths. A detailed description of these benchmarks can be found in Appendix A.

Table 1: Experimental results on Standard & Long CL benchmarks with *t5-large* and *Llama-2-7b-chat*. **Bold** and underlined numbers denote the best and second-best results, respectively.

| Method | **Standard** ($N = 4$) | | | | **Long** ($N = 15$) | | | |
|---|---|---|---|---|---|---|---|---|
| | Order1 | Order2 | Order3 | Acc | Long1 | Long2 | Long3 | Acc |
| | | | | *google-t5/t5-large* | | | | |
| SeqLoRA | 25.7 | 24.0 | 35.2 | $28.3_{\pm6.0}$ | 12.3 | 10.1 | 10.1 | $10.8_{\pm1.3}$ |
| EWC(Kirkpatrick et al., 2017) | 48.7 | 47.7 | 54.5 | $50.3_{\pm3.7}$ | 45.3 | 44.5 | 45.6 | $45.1_{\pm0.6}$ |
| LwF(Li & Hoiem, 2017) | 54.4 | 53.1 | 49.6 | $52.3_{\pm2.5}$ | 50.1 | 43.1 | 47.4 | $46.9_{\pm3.5}$ |
| L2P(Wang et al., 2022b) | 60.3 | 61.7 | 61.1 | $60.7_{\pm0.7}$ | 57.5 | 53.8 | 56.9 | $56.1_{\pm2.0}$ |
| IncLoRA | 68.6 | 59.7 | 75.0 | $67.8_{\pm7.7}$ | 60.3 | 60.5 | 53.2 | $58.0_{\pm4.2}$ |
| MIGU(Du et al., 2024) | 77.2 | 76.7 | 75.4 | $76.4_{\pm0.9}$ | 71.3 | 67.7 | 67.3 | $68.7_{\pm2.2}$ |
| O-LoRA(Wang et al., 2023b) | 74.9 | 73.4 | 75.6 | $74.6_{\pm1.1}$ | 71.5 | 66.7 | 71.3 | $69.8_{\pm2.7}$ |
| SD-LoRA(Wu et al., 2025) | 67.7 | 55.9 | 60.9 | $61.5_{\pm5.9}$ | 69.3 | 69.8 | 70.0 | $69.7_{\pm0.4}$ |
| MoCL(Wang et al., 2024b) | 75.6 | 75.4 | 76.7 | $75.9_{\pm0.7}$ | 69.6 | 70.2 | 70.9 | $70.2_{\pm0.7}$ |
| AM-LoRA(Liu et al., 2024) | 78.1 | **79.8** | 76.2 | $78.0_{\pm1.8}$ | 72.7 | 73.3 | 71.8 | $72.6_{\pm0.8}$ |
| **PS-LoRA** | **80.0** | 79.1 | **79.6** | $\mathbf{79.6_{\pm0.5}}$ | **74.2** | **76.5** | **75.7** | $\mathbf{75.5_{\pm1.2}}$ |
| | | | | *meta-llama/Llama-2-7b-chat* | | | | |
| SeqLoRA | 73.4 | 75.6 | 75.5 | $74.8_{\pm1.2}$ | 69.0 | 70.5 | 66.9 | $68.8_{\pm1.8}$ |
| IncLoRA | 75.9 | 72.6 | 76.8 | $75.1_{\pm2.2}$ | 70.7 | 70.8 | 69.2 | $70.2_{\pm0.9}$ |
| MIGU(Du et al., 2024) | 77.7 | 77.1 | 78.9 | $77.9_{\pm0.9}$ | 71.2 | 70.6 | 70.5 | $70.5_{\pm0.4}$ |
| OLoRA(Wang et al., 2023b) | 76.8 | 75.7 | 75.7 | $76.0_{\pm0.6}$ | 71.1 | 68.9 | 73.8 | $71.3_{\pm2.5}$ |
| SD-LoRA (Wu et al., 2025) | 76.6 | 74.5 | 76.8 | $76.0_{\pm1.3}$ | 70.2 | 68.4 | 70.9 | $69.8_{\pm1.3}$ |
| MoCL(Wang et al., 2024b) | 78.4 | 77.7 | 78.4 | $78.2_{\pm0.4}$ | 75.2 | 70.7 | 74.8 | $73.6_{\pm2.5}$ |
| **PS-LoRA** | **80.9** | **81.2** | **80.4** | $\mathbf{80.8_{\pm0.4}}$ | **76.7** | **76.1** | **76.2** | $\mathbf{76.3_{\pm0.3}}$ |

**Backbones.** For the Standard & Long benchmark, we follow O-LoRA (Wang et al., 2023b) to evaluate both encoder-decoder (*T5-Large* (Raffel et al., 2020)) and decoder-only (*LLaMA-2-7B* (Touvron et al., 2023)) models. For TRACE generation tasks, we include three LLM backbones: *Mistral-7B-Instruct-v0.3*, *LLaMA-2-7B*, and *Gemma-2B-it*. In addition, for ViT tasks we follow SD-LoRA (Wu et al., 2025) and use *ViT-B/16* (Dosovitskiy et al., 2020) as the backbone. Please see Appendix A.2 and C.2 for the definition of the **evaluation metrics** and the **baseline** details.

## 4.1 EXPERIMENTAL RESULTS

**Results on Standard & Long Benchmarks**. To evaluate effectiveness under varying task lengths and orders, we conduct experiments on the **Standard** (4 tasks) and **Long** (15 tasks) benchmarks with two LLM backbones. As shown in Table 1, the long benchmark yields lower performance, reflecting the increased sequence length and cumulative task interference. Nevertheless, PS-LoRA delivers consistent gains, outperforming AM-LoRA by 1.6% and 2.9% on the Standard

Table 2: Comparison of FR, FWT, and BWT on the Standard & Long benchmarks.

| Method | **Standard** ($N = 4$) | | | **Long** ($N = 15$) | | |
|---|---|---|---|---|---|---|
| | FR↓ | FWT↑ | BWT↑ | FR↓ | FWT↑ | BWT↑ |
| SeqLoRA | 67.82 | -0.29 | -67.82 | 70.05 | -5.65 | -69.19 |
| IncLoRA | 5.07 | -0.46 | -5.00 | 13.54 | -9.07 | -10.37 |
| O-LoRA | 3.29 | -0.46 | -3.26 | 10.00 | -5.24 | -7.03 |
| SD-LoRA | 3.06 | -2.06 | -2.87 | 9.12 | -4.02 | -8.17 |
| MoCL | 4.14 | -2.37 | -2.11 | 10.25 | **-2.60** | -7.89 |
| **PS-LoRA** | **1.99** | **0.01** | **-1.84** | **6.32** | -3.13 | **-0.68** |

and Long benchmarks, respectively. Across architectures, PS-LoRA shows consistent improvements, achieving up to 5.7% gains over baselines on *T5-Large* and up to 5.0% on *LLaMA-2-7B*. Beyond overall accuracy, we analyze other CL metrics (i.e., *FR*, *FWT*, *BWT*) to assess forgetting and knowledge transfer (see Table 2). PS-LoRA achieves the lowest FR on both Standard (1.99%) and Long (6.32%) benchmarks, showing strong resistance to catastrophic forgetting, and it maintains competitive *FWT* and *BWT*. The consistent gains across continual learning metrics demonstrate robustness in both short and long horizons, evidencing our PS-LoRA's effectiveness at reducing forgetting.

To better demonstrate the performance of PS-LoRA across different task lengths, we plot the test accuracy after completing each task, as shown in Fig. 7(a)(b). Compared to other methods, PS-LoRA exhibits significantly smaller performance fluctuations and consistently maintains strong performance throughout the training process. This stable behavior highlights its strong ability to resist catastrophic forgetting and adapt to new tasks without sacrificing previous knowledge.

**Results on Computer Vision Tasks.** As shown in Table 3, to evaluate the generalizability of PS-LoRA beyond NLP tasks, we consider two widely used class-incremental learning vision benchmarks

Table 3: Experimental results on ImageNet-R with varying task lengths.

| Method | IN-R ($N=5$) Acc | AAA | IN-R ($N=10$) Acc | AAA | IN-R ($N=20$) Acc | AAA |
|---|---|---|---|---|---|---|
| Full FT | 64.92 | 75.57 | 60.57 | 72.31 | 49.95 | 65.32 |
| L2P (Wang et al., 2022b) | 73.04 | 76.94 | 71.26 | 76.13 | 68.97 | 74.16 |
| DualPrompt (Wang et al., 2022a) | 69.99 | 72.24 | 68.22 | 73.81 | 65.23 | 71.30 |
| HiDe-Prompt (Wang et al., 2023a) | 74.77 | 78.15 | 74.65 | 78.46 | 73.59 | 77.93 |
| SD-LoRA(Wu et al., 2025) | 79.15 | 83.01 | **77.34** | 82.04 | 75.26 | 80.22 |
| **PS-LoRA** | **79.68** | **83.82** | 77.15 | **82.12** | **75.35** | **80.50** |

Table 4: Experimental results on the TRACE benchmark with varying LLM backbones.

| Method | mistralai / Mistral-7B-Instruct-v0.3 AAA ↑ | BWT ↑ | meta-llama / LLaMA-2-7B-Chat AAA ↑ | BWT ↑ | google / Gemma-2B-it AAA ↑ | BWT ↑ |
|---|---|---|---|---|---|---|
| SeqLoRA | $46.94_{\pm1.2}$ | $-11.41_{\pm0.6}$ | $34.30_{\pm1.2}$ | $-18.50_{\pm0.8}$ | $31.89_{\pm0.8}$ | $-15.28_{\pm0.4}$ |
| EWC | $52.45_{\pm1.3}$ | $-5.98_{\pm0.8}$ | $42.36_{\pm1.2}$ | $-5.97_{\pm0.8}$ | $28.35_{\pm1.6}$ | $-16.96_{\pm1.2}$ |
| L2P | $49.32_{\pm0.8}$ | $-5.34_{\pm0.6}$ | $36.23_{\pm0.8}$ | $-8.25_{\pm0.8}$ | $31.14_{\pm1.2}$ | $-15.77_{\pm0.7}$ |
| DualPrompt | $51.14_{\pm1.2}$ | $-6.13_{\pm0.5}$ | $37.69_{\pm1.2}$ | $-8.03_{\pm0.8}$ | $32.42_{\pm1.0}$ | $-14.25_{\pm0.5}$ |
| HiDeLoRA | $51.81_{\pm0.9}$ | $-6.25_{\pm0.3}$ | $41.60_{\pm0.8}$ | $-7.12_{\pm0.4}$ | $33.25_{\pm0.9}$ | $-13.66_{\pm0.5}$ |
| O-LoRA | $52.02_{\pm0.8}$ | $-8.13_{\pm0.6}$ | $42.78_{\pm0.8}$ | $-7.16_{\pm0.4}$ | $33.73_{\pm0.8}$ | $-12.36_{\pm0.4}$ |
| TreeLoRA | $54.77_{\pm1.1}$ | $\mathbf{-3.77_{\pm0.4}}$ | $43.52_{\pm1.0}$ | $-3.46_{\pm0.4}$ | $33.41_{\pm0.9}$ | $-8.50_{\pm0.5}$ |
| **PS-LoRA** | $\mathbf{54.95_{\pm0.8}}$ | $\underline{-4.02_{\pm0.5}}$ | $\mathbf{45.50_{\pm0.9}}$ | $\mathbf{-3.24_{\pm0.4}}$ | $\mathbf{35.80_{\pm1.1}}$ | $\mathbf{-6.79_{\pm0.5}}$ |

ImageNet-R under varying task lengths ($N = 5, 10, 20$). We integrate LoRA modules into ViT-B/16 and apply our method across sequential tasks. While PS-LoRA matches the SOTA SD-LoRA on vision benchmarks, it exceeds SD-LoRA on NLP tasks and simultaneously curbs catastrophic forgetting. All these experiments demonstrate that PS-LoRA's gains extend beyond NLP and validate its robust cross-modal generalizability. **Regarding results on the TRACE benchmark**, table 4 presents the performance across the more challenging multilingual tasks, such as code generation, and mathematical reasoning. It can be seen that our PS-LoRA consistently achieves competitive or superior performance across different backbones. For example, PS-LoRA outperforms all methods by at least $2.0\%$ and $2.4\%$ on *Llama-2-7B* and *Gemma-2B-it*, respectively. These results indicate that PS-LoRA can generalize effectively to complex NLP tasks, further demonstrating robust generalization beyond the training distribution.

**Ablation Studies on PS-LoRA Components.** We ablate two components: (1) *Parameter Stability loss*, which aligns the signs of current-task weights with the accumulated task vector; and (2) *Merging strategies*, which reuse prior model knowledge. All experiments follow the main setup. As shown in Table 5, removing either component clearly degrades performance, confirming their complementary roles in mitigating forgetting and stabilizing learning. Removing the stability loss causes the largest drop, indicating that sign alignment helps prevent conflicting updates. Replacing the magnitude-based merging with simple addition weakens knowledge consolidation and increases forgetting. Using both components yields the best results, highlighting the importance of controlling update direction and reusing historical parameters in continual NLP learning.

## 4.2 DISCUSSION

In addition to the improvements shown in Sec. 4.1, we further analyze its underlying behavior and characteristics. In the following, we address several key questions to provide more insights into how and why PS-LoRA works effectively in the continual learning setting.

**Q-1: Why does CL often lead to the abrupt performance drop and the large parameter distribution shift observed in Fig. 1?** Using Long2 as an example, we apply MDS (Kruskal, 1964b;a) to visualize the first four tasks (Fig. 6).Task $\mathcal{T}_4$ departs markedly from the first three, indicating low similarity and a pronounced distribution shift. Under the SD-LoRA view (Wu et al., 2025), CL seeks a shared low-loss region; a sharp shift like $\mathcal{T}_4$ drives the model toward a task-specific optimum, triggering abrupt updates that disrupt prior

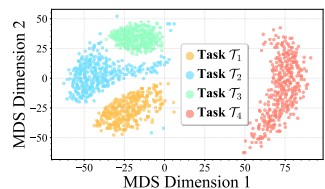

Figure 6: Feature visualization across tasks.

Table 5: Ablation study of different components of PS-LoRA on T5-large. Results (%) are averaged over three random task orders on two continual learning benchmarks.

| PS-Loss | Merging | Standard ($N = 4$) | | | | Long ($N = 15$) | | | |
|---|---|---|---|---|---|---|---|---|---|
| | | Order1 | Order2 | Order3 | avg | Long1 | Long2 | Long3 | avg |
| ✗ | ✗ | 68.6 | 59.7 | 75.0 | $67.8_{\pm7.7}$ | 60.3 | 60.5 | 53.2 | $58.0_{\pm4.2}$ |
| ✗ | ✓ | 76.9 | 74.4 | 77.0 | $76.1_{\pm1.5}$ | 70.9 | 69.8 | 70.7 | $70.5_{\pm0.6}$ |
| ✓ | ✗ | 79.2 | 78.3 | 78.3 | $78.6_{\pm0.5}$ | 72.9 | 74.7 | 73.1 | $73.6_{\pm1.0}$ |
| ✓ | ✓ | **80.0** | **79.1** | **79.6** | $\mathbf{79.6}_{\pm0.5}$ | **74.2** | **76.5** | **75.7** | $\mathbf{75.5}_{\pm1.2}$ |

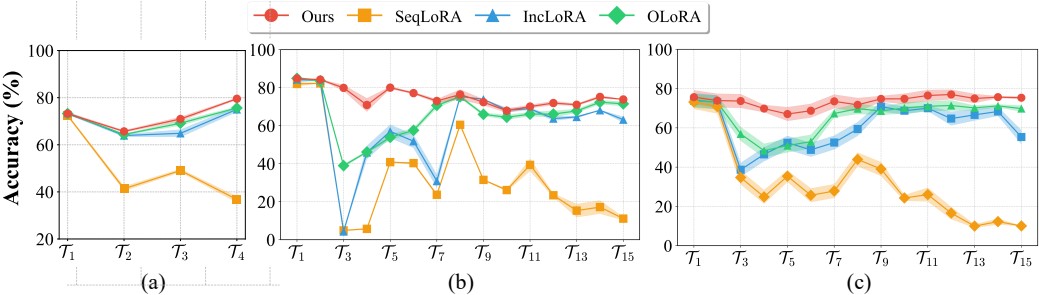

Figure 7: Test accuracy across different tasks. (a-b): Standard and Long benchmarks. (c) shows average accuracy over three task orders on Long (order sensitivity). More orders are in Appendix C.3.

knowledge and yield large parameter shifts. To mitigate this, PS-LoRA explicitly limits update magnitudes, preserving prior knowledge and reducing catastrophic forgetting. Results across different task orders (Fig. 7c) show consistently stable performance, indicating reduced interference from dissimilar tasks and improved order robustness in NLP continual learning.

**Q-2: Does the proposed parameter stability loss effectively alleviate the issue of large updates with opposite signs flipping parameter directions?** To verify whether the proposed PS-loss effectively mitigates the problem of large updates with opposite signs, we analyze the proportion of updates that align or misalign in sign

Table 6: PS-LoRA w/o Merging helps avoid sign flips in parameter updates.

| Task | IncLoRA | | | PS-LoRA w/o Merging | | |
|---|---|---|---|---|---|---|
| | Same | Opposite | Acc | Same | Opposite | Acc |
| $\mathcal{T}_2$ | 49.16% | 9.74% | 73.9 | 63.22% | 1.30% | 74.4 |
| $\mathcal{T}_3$ | 51.61% | 5.72% | 68.9 | 61.59% | 1.07% | 71.7 |
| $\mathcal{T}_4$ | 49.84% | 22.30% | 39.2 | 60.52% | 2.17% | 56.0 |

with the accumulated LoRAs. As shown in Table 6, the introduction of Parameter Stability loss dramatically reduces the ratio of sign-inconsistent updates from 22.3% to 2.17%, indicating that updates become more sign-consistent. This improvement enhances knowledge retention and yields substantial accuracy gains.

**Q-3: Can our proposed PS-LoRA be combined with existing orthogonality constraints to further enhance performance?** As shown in Table 7, integrating our PS-LoRA with O-LoRA leads to a clear performance improvement over using O-LoRA alone. This result suggests that our method is complementary to orthogonality-based approaches rather than conflicting with them. Moreover, these findings highlight the practical utility of our PS-LoRA in facilitating stable knowledge accumulation and alleviating forgetting.

Table 7: Performance of O-LoRA increases when combined with PS-Loss and PS-LoRA.

| Method | Standard ($N = 4$) | | | Long ($N = 15$) | | |
|---|---|---|---|---|---|---|
| | Order1 | Order2 | Order3 | Long1 | Long2 | Long3 |
| O-LoRA | 74.9 | 73.4 | 75.6 | 71.5 | 66.7 | 71.3 |
| +PS-Loss | 79.3 | 78.1 | 79.4 | 76.2 | 75.1 | 76.7 |
| +PS-LoRA | 79.4 | 79.6 | 79.2 | 74.3 | 77.2 | 76.2 |

## 5 CONCLUSION

In this work, we identify an empirical phenomenon where abrupt performance drops correlate strongly with significant shifts in parameter distribution during CL. A deeper analysis reveals that updates with sign changes are a key factor causing forgetting. Motivated by this insight, we propose the Parameter Stability Loss to explicitly constrain such sign-flipping updates and mitigate catastrophic forgetting. In addition, we integrate a post-training magnitude-based merging strategy that bridges earlier directions with the current one and further combats the inevitable drift toward new tasks without incurring extra training costs. Extensive experiments across varying datasets, task lengths and diverse backbone architectures demonstrate the consistent effectiveness of our PS-LoRA framework.

ETHICS STATEMENT

This research adheres to the ICLR Code of Ethics. Our work does not involve human subjects, personally identifiable information, or sensitive data. All datasets used in this study are publicly available and have been released by their original authors with appropriate licenses. We are not aware of any privacy, fairness, or security concerns directly arising from the methodology or results. The authors take responsibility for ensuring that the work complies with ethical research standards, including research integrity, data handling, and reproducibility.

REPRODUCIBILITY STATEMENT

We have taken steps to ensure the reproducibility of our results. The architecture, training procedures, and evaluation protocols are described in detail in Sections 4. Hyper-parameters and implementation details are provided in Appendix C.1. All datasets used in the experiments are standard benchmarks with publicly available access. To further facilitate reproducibility, we will release the source code and instructions for reproducing all experiments upon acceptance.

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

# Contents

## A    BENCHMARKS

### A.1    DATASETS

**Standard & Long**   Table 8 presents the detailed statistics of the 15 datasets utilized in our continual learning (CL) experiments, along with their corresponding evaluation metrics. These datasets are primarily drawn from established CL benchmarks (Zhang et al., 2016), as well as the GLUE (Wang et al., 2018) and SuperGLUE (Wang et al., 2019) benchmarks. Additionally, we include the IMDB movie review dataset, following the experimental setup of ProgPrompt(Razdaibiedina et al., 2023) and O-LoRA((Wang et al., 2023b)).

**TRACE**   (Wang et al., 2023c) is a benchmark designed especially for the continual learning with LLMs. It consists of 8 distinct datasets spanning challenging tasks, including domain-specific tasks, multilingual capabilities, code generation, and mathematical reasoning, with each task containing $5,000$ instances. Specifically, the eight tasks are: **C-STANCE, FOMC, MeetingBank, Py150, ScienceQA, NumGLUE-cm, NumGLUE-ds, and, 20Minuten**. All datasets are standardized into a unified format, allowing for effortless automatic evaluation of LLMs. Therefore, it contains a total of $200,000$ samples, with $40,000$ training examples and $16,000$ testing examples.

Note that TRACE contains a wide range of different tasks, including domain-specific tasks, multilingual capabilities, code generation, and mathematical reasoning. The performance measure for each task is different: For C-STANCE and FOMC tasks, we use accuracy as the evaluation metric to assess the model's classification performance. MeetingBank task performance is evaluated using the ROUGE-L score, which measures the longest common subsequence between the generated and reference summaries. For code-related task Py150, we employ a similarity score to evaluate the quality of generated code. The ScienceQA task is evaluated using accuracy to measure the correctness of scientific question answering. Both NumGLUE-cm and NumGLUE-ds tasks, which focus on mathematical reasoning, use accuracy as their evaluation metrics. Lastly, for the multilingual task 20Minuten, we utilize the SARI score to assess the quality of text simplification.

**Computer Vision**   ImageNet-R consists of 200 ImageNet classes (Deng et al., 2009) rendered in artistic styles. As common practices (Wu et al., 2025), we split ImageNet-R into 5/10/20 tasks (40/20/10 classes per task).

## A.2 METRICS

**Accuracy** For evaluation metric, we adopt the Average Accuracy (Acc). Formally, let $a_{i,j}$ denote the test accuracy on the $i$-th task $\mathcal{T}_i$ after training on $\mathcal{T}_j$. Then the average accuracy of all seen tasks can be defined as,

$$\text{Acc} = \frac{\sum_{i=1}^{N} |\mathcal{D}_i| \cdot a_{i,N}}{\sum_{i=1}^{N} |\mathcal{D}_i|},$$

where $|\mathcal{D}_i|$ refers to the number of test samples in task $\mathcal{T}_i$.

**Continual Learning Metrics** We adopt the notation $a_{i,N}$ is the final accuracy on task $T_i$ after learning all $N$ tasks. The metrics are defined as follows:

**Backward Transfer (BWT)**

$$\text{BWT} = \frac{1}{N-1} \sum_{i=1}^{N-1} (a_{i,N} - a_{i,i})$$

This measures the change in performance on task $T_i$ from immediately after its learning to after learning all tasks; negative values indicate forgetting.

**Forward Transfer (FWT)**

$$\text{FWT} = \frac{1}{N-1} \sum_{i=2}^{N} (a_{i,i-1} - a_{\text{scratch},\, i})$$

Here, $a_{\text{scratch},\, i}$ is the accuracy when training task $T_i$ from scratch, assessing how prior tasks positively or negatively influence new task learning.

**Forgetting Rate (FR)**

$$\text{FR} = \frac{1}{N-1} \sum_{i=1}^{N-1} \left( \max_{j \leq i} a_{i,j} - a_{i,N} \right)$$

This captures how much accuracy on each task $T_i$ decreases from its peak (during training) to the final performance after all tasks.

Table 8: The details of 15 datasets used in CL experiments. NLI denotes natural language inference, QA denotes questions and answers task. First five tasks correspond to the standard CL benchmark, all other tasks are used in long-sequence experiments.

| Dataset name | Category | Task | Domain |
|---|---|---|---|
| 1. Yelp | CL Benchmark | sentiment analysis | Yelp reviews |
| 2. Amazon | CL Benchmark | sentiment analysis | Amazon reviews |
| 3. DBpedia | CL Benchmark | topic classification | Wikipedia |
| 4. Yahoo | CL Benchmark | topic classification | Yahoo Q&A |
| 5. AG News | CL Benchmark | topic classification | news |
| 6. MNLI | GLUE | NLI | various |
| 7. QQP | GLUE | paragraph detection | Quora |
| 8. RTE | GLUE | NLI | news, Wikipedia |
| 9. SST-2 | GLUE | sentiment analysis | movie reviews |
| 10. WiC | SuperGLUE | word sense disambiguation | lexical databases |
| 11. CB | SuperGLUE | NLI | various |
| 12. COPA | SuperGLUE | QA | blogs, encyclopedia |
| 13. BoolQA | SuperGLUE | boolean QA | Wikipedia |
| 14. MultiRC | SuperGLUE | QA | various |
| 15. IMDB | SuperGLUE | sentiment analysis | movie reviews |

## A.3 TASK SEQUENCE ORDERS

The task sequences employed in our CL experiments for both T5 and LLaMA models are summarized in Table 9. Order 1-3 correspond to the standard CL benchmark adopted by prior works. Long 1-3

are long-sequence orders spanning 15 tasks, following ProgPrompt(Razdaibiedina et al., 2023) and O-LoRA((Wang et al., 2023b)).

Table 9: Six different orders of task sequences used for continual learning experiments.

| Order | Model | Task Sequence |
|---|---|---|
| order1 | T5, LLaMA | dbpedia → amazon → yahoo → ag |
| order2 | T5, LLaMA | dbpedia → amazon → ag → yahoo |
| order3 | T5, LLaMA | yahoo → amazon → ag → dbpedia |
| long1 | T5, LLaMA | mnli → cb → wic → copa → qqp → boolqa → rte → imdb → yelp → amazon → sst-2 → dbpedia → ag → multirc → yahoo |
| long2 | T5, LlaMA | multirc → boolqa → wic → mnli → cb → copa → qqp → rte → imdb → sst-2 → dbpedia → ag → yelp → amazon → yahoo |
| long3 | T5, LlaMA | yelp → amazon → mnli → cb → copa → qqp → rte → imdb → sst-2 → dbpedia → ag → yahoo → multirc → boolqa → wic |

## A.4 TASK INSTRUCTIONS

Table 10 presents the prompt templates used across various tasks. Specifically, natural language inference (NLI) tasks include MNLI, RTE, and CB; sentiment classification (SC) comprises Amazon, Yelp, SST-2, and IMDB; while topic classification (TC) includes AG News, DBpedia, and Yahoo Answers.

Table 10: Instructions for different tasks.

| Task | Prompts |
|---|---|
| NLI | What is the logical relationship between the "sentence 1" and the "sentence 2"? Choose one from the option. |
| QQP | Whether the "first sentence" and the "second sentence" have the same meaning? Choose one from the option. |
| SC | What is the sentiment of the following paragraph? Choose one from the option. |
| TC | What is the topic of the following paragraph? Choose one from the option. |
| BoolQA | According to the following passage, is the question true or false? Choose one from the option. |
| MultiRC | According to the following passage and question, is the candidate answer true or false? Choose one from the option. |
| WiC | Given a word and two sentences, whether the word is used with the same sense in both sentence? Choose one from the option. |

## B PARAMETER SHIFT DISTRIBUTIONS IN LONG TASK SEQUENCES

This section provides the *Parameter Shift Distributions* under three long-task orders in Fig.8, Fig.9, Fig.10 respectively. For each setting, we visualize the distribution of selected parameters after training with both **Incremental LoRA** and **our method**. Our analysis focuses on the decoder block weights in the T5 model.

For the $\mathcal{T}_i$ task, we compute the cumulative LoRA updates by summing over all previous low-rank adapters:

$$\Delta \mathbf{W}_i = \sum_{j=1}^{i} \mathbf{A}_j \mathbf{B}_j,$$

where $\mathbf{A}_j$ and $\mathbf{B}_j$ denote the low-rank matrices of the $\mathcal{T}_j$ task's LoRA adapter.

Then, we identify the top 20% of parameters with the largest absolute values in $\Delta \mathbf{W}_i$ as the *important parameters*, and we perform average pooling on the parameters to enhance their feature representation. Finally we plot their value distributions across tasks to analyze shift patterns.

Generally, our approach results in a more stable parameter distribution, indicating enhanced robustness and less catastrophic forgetting.

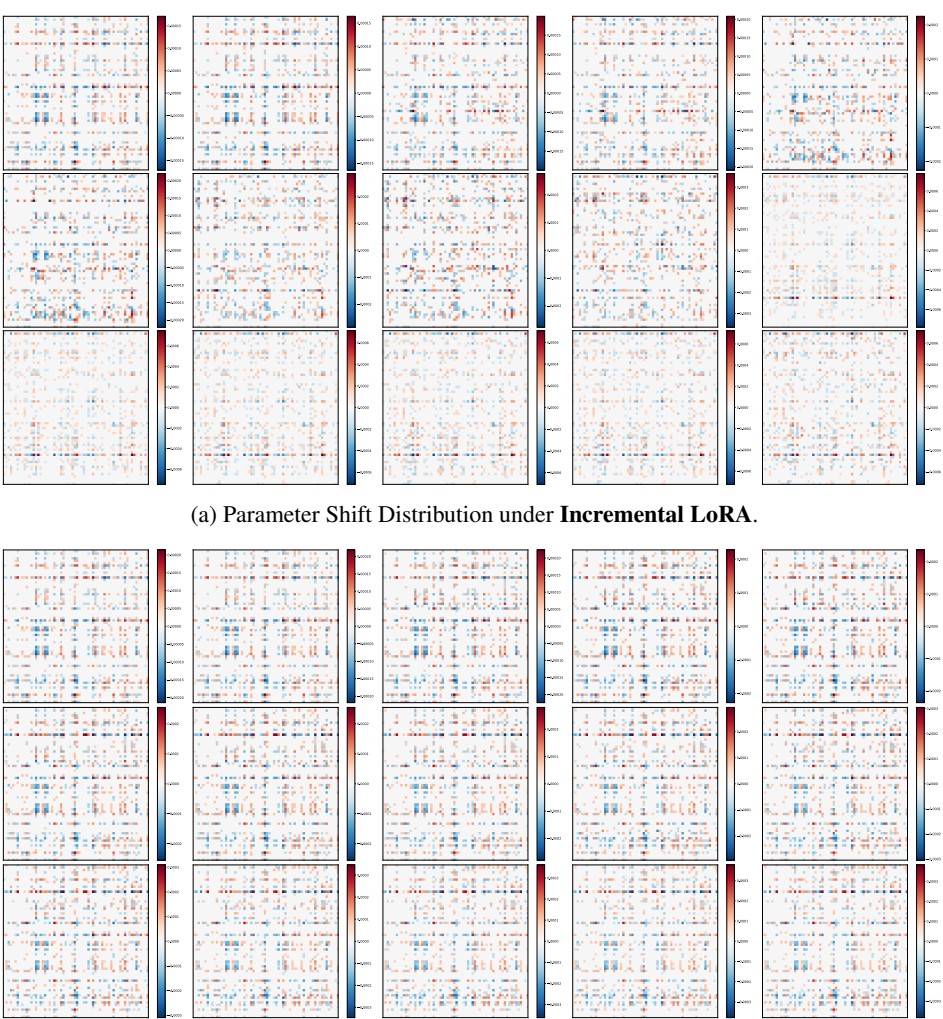

(a) Parameter Shift Distribution under **Incremental LoRA**.

(b) Parameter Shift Distribution under **Our Method**.

Figure 8: Comparison of parameter shift distributions for **Long1** under different methods. Our method shows more consistent parameter evolution and reduced directional conflict across tasks.

## C EXPERIMENTS DETAILS

### C.1 IMPLEMENTING DETAILS

All experiments were conducted using two NVIDIA RTX 4090 GPUs (40GB each) with DeepSpeed-enabled distributed training. Our method is implemented based on the training framework provided by O-LoRA under the MIT License.

We insert LoRA adapters into the query and value projection matrices of all Transformer layers, with each adapter configured to have a rank of $r = 8$, a dropout rate of $0.1$, and a scaling factor of $1$. For the t5-large model, we use a learning rate of $0.001$ and a batch size of $8$, training each task for one epoch. The parameter stability loss coefficient $\lambda$ is set to $0.1$ for long tasks and $0.001$ for

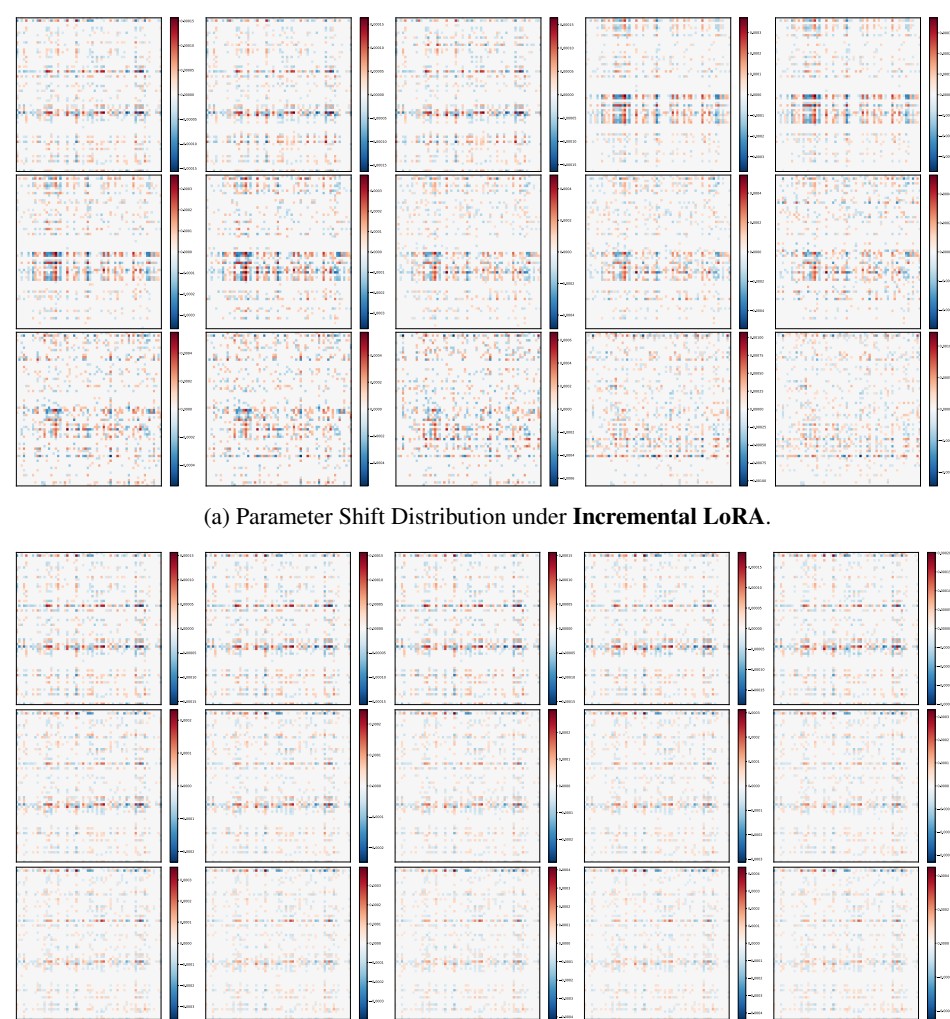

(a) Parameter Shift Distribution under **Incremental LoRA**.

(b) Parameter Shift Distribution under **Our Method**.

Figure 9: Comparison of parameter shift distributions for **Long2** under different methods. Our method shows more consistent parameter evolution and reduced directional conflict across tasks.

standard sequences. For the `llama2-7b` model, we use a learning rate of $0.0003$ and a batch size of $4$, also training each task for one epoch, with the parameter stability loss coefficient set to $0.01$ for long tasks and $0.001$ for standard sequences. We conducted detailed hyperparameter sensitivity experiments in Tab. 13 to clarify our choice. In all experiments using O-LoRA, the orthogonal loss coefficient is fixed at $\lambda = 0.5$. When applying neuron merge, we scale the orthogonal component by a factor of 3, and parallel component by a factor of 1. We report results based on three random seeds: 42, 1121, and 3407.

## C.2 BASELINES

- **Zero-shot**: directly tests pretrained model on benchmarks without any finetuning.

- **SeqLoRA**: assigns one LoRA for all tasks, and sequentially finetuning this LoRA on each task.

- **Replay**: This method mitigates forgetting by maintaining a fixed-size memory buffer that stores a subset of past samples. During training on a new task, both the current task data and replayed samples from earlier tasks are jointly used to fine-tune the model.

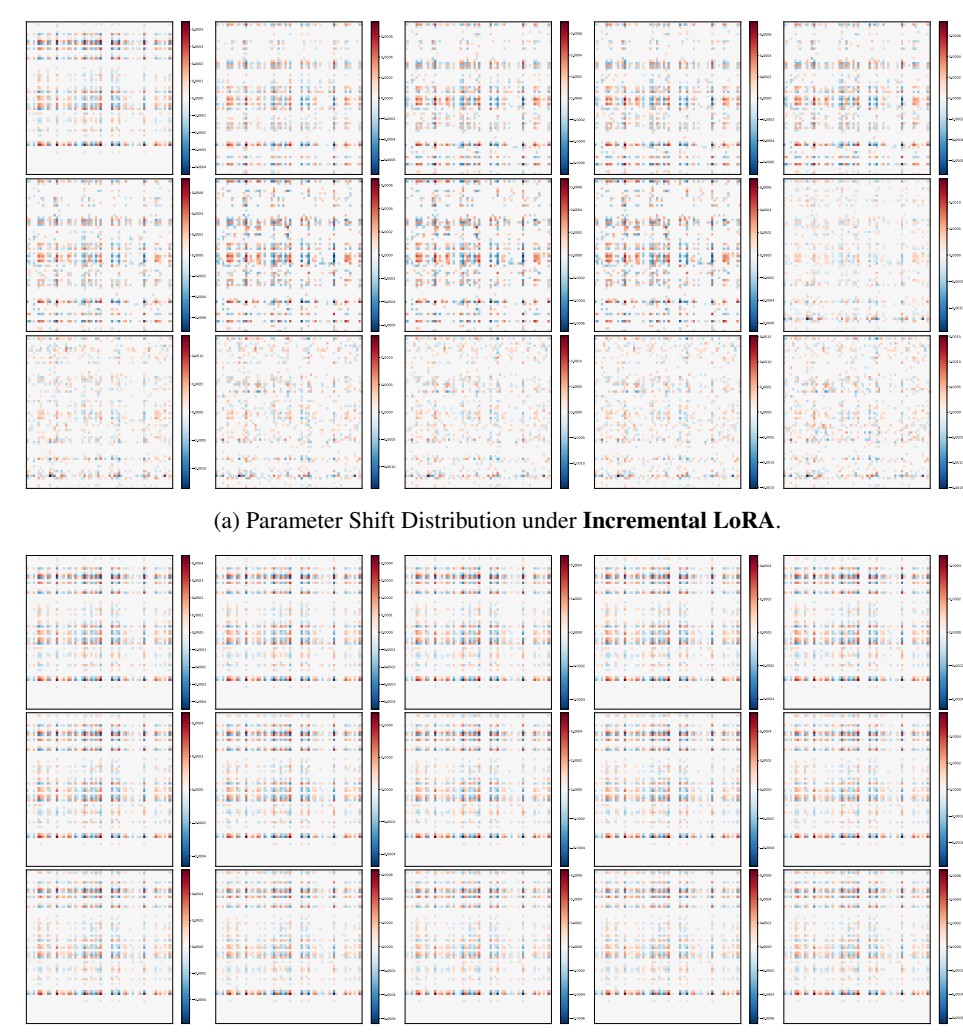

(a) Parameter Shift Distribution under **Incremental LoRA**.

(b) Parameter Shift Distribution under **Our Method**.

Figure 10: Comparison of parameter shift distributions for **Long3** under different methods. Our method shows more consistent parameter evolution and reduced directional conflict across tasks.

- **EWC** (Kirkpatrick et al., 2017): Elastic Weight Consolidation imposes a quadratic penalty on parameter updates, discouraging changes to weights that are crucial to previously learned tasks, based on their estimated importance derived from the Fisher Information Matrix.

- **LwF** (Li & Hoiem, 2017): Learning without Forgetting avoids storing old data by preserving responses of the shared representation on past tasks via a distillation loss. This helps maintain stable internal representations when adapting to new tasks.

- **L2P** (Wang et al., 2022b): Learning to Prompt introduces a pool of learnable prompts and selects task-relevant prompts for each input dynamically. This instance-wise prompt retrieval enables the model to adapt without modifying the pretrained backbone.

- **LFPT5** (Huang et al., 2021): A prompt-based continual learner built on T5, which jointly optimizes soft prompts for task solving and sample generation. The generated pseudo-examples are then utilized in a rehearsal-like manner to retain previous knowledge.

- **IncLoRA**: IncLoRA incrementally adds a task-specific LoRA module per task and keeps previously learned modules frozen. Each task maintains its dedicated adapter.

- **MIGU** (Du et al., 2024): MIGU selectively updates gradient of parameters only with magnitude above threshold, supposing magnitude distribution among different tasks distinguishes

them from each other. It can be added to different architecture. Since out method is based on IncLoRA, we choose IncLoRA+MIGU as our baseline.

- **O-LoRA**(Wang et al., 2023b): O-LoRA bases on IncLoRA framework, while it imposes an orthogonal regularization to restrict the update of parameters in subspace.

- **LB-CL**(Qiao & Mahdavi, 2024): LB-CL also bases on IncLoRA framework, it initializes new LoRA with SVD decomposition of previous task parameters and enforces orthogonality across task subspaces via gradient projection.

- **MoCL**(Wang et al., 2024b): MoCL calculats task similairty coefficient and dynamically combines trained LoRAs in order to eliminate forgetting.

- **AM-LoRA**(Liu et al., 2024): AM-LoRA bases on IncLoRA and adaptively integrates their knowledge using an attention mechanism with L1 sparsity constraints.

- **PerTaskFT**: trains a separate LoRA model for each task.

- **MTL**: trains a model on all tasks as multi-task learning, serving as the benchmark's upper bound of the performance limit.

- **PS-LoRA**: Our method trains model with Parameter Stability Loss and magnitude-selected mergeing strategy.

## C.3 TASK ACCURACY

In Fig.11, We provide additional sequential cases similar to this Fig.7, further validating the robustness of the PS-LoRA method.

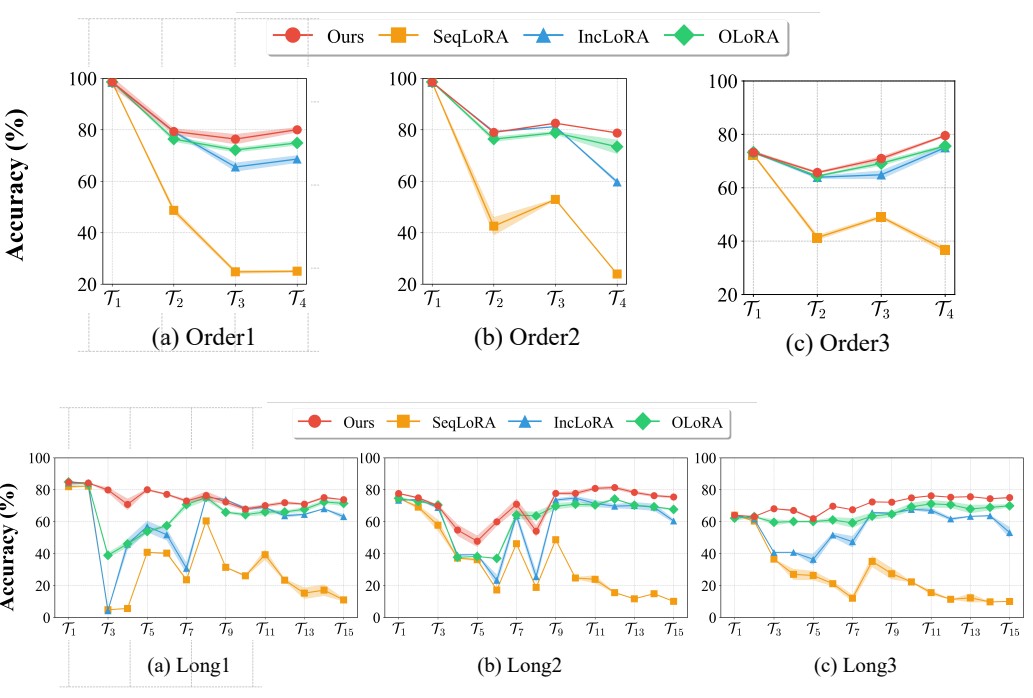

Figure 11: The average accuracy for each sequence with incremental tasks

# D SUPPLEMENTARY EXPERIMENTS

## D.1 EXPLORATION OF SAME SIGN PERFORMANCE

In this experiment, we adopt an incremental LoRA training strategy where a new trainable LoRA module $\mathbf{A}_t\mathbf{B}_t$ is assigned for each incoming task $\mathcal{T}_t$. During training, no constraints are imposed on

Table 11: Performance of experiments that manually add sign mask after each task

| Method | Standard ($N = 4$) | | | | Long ($N = 15$) | | | |
|---|---|---|---|---|---|---|---|---|
| | Order1 | Order2 | Order3 | avg | Long1 | Long2 | Long3 | avg |
| IncLoRA | 68.6 | 59.7 | 75.0 | 67.8 | 60.3 | 60.5 | 53.2 | 58.0 |
| Decomposition | 77.2 | 72.9 | 72.4 | 74.1 | 67.1 | 68.4 | 68.8 | 68.1 |
| **Ours** | **79.2** | **78.3** | **78.3** | **78.6** | **72.9** | **74.7** | **73.1** | **73.6** |

the parameter updates, allowing full flexibility for task-specific adaptation. After training on a task is completed, we retain only the components of the current LoRA whose signs are consistent with the element-wise sum of all previously learned LoRA modules $\sum_{i=1}^{t-1} \mathbf{A}_i \mathbf{B}_i$. This consistency check helps mitigate interference with previously acquired knowledge. The resulting sign-consistent matrix is then subjected to Singular Value Decomposition (SVD), and the low-rank factors from the decomposition are used to construct the LoRA module for the current task. This process enables continual learning by progressively integrating task-specific knowledge while controlling for conflicting parameter directions across tasks. Results are shown in Table 11.

## D.2 DIFFERENT MERGING PERFORMANCE ON TASKS

Specifically, for the long-order setting, we compute the change in average accuracy for each task before and after merging, evaluated after training the final task. A corresponding heatmap is shown in Fig.12.

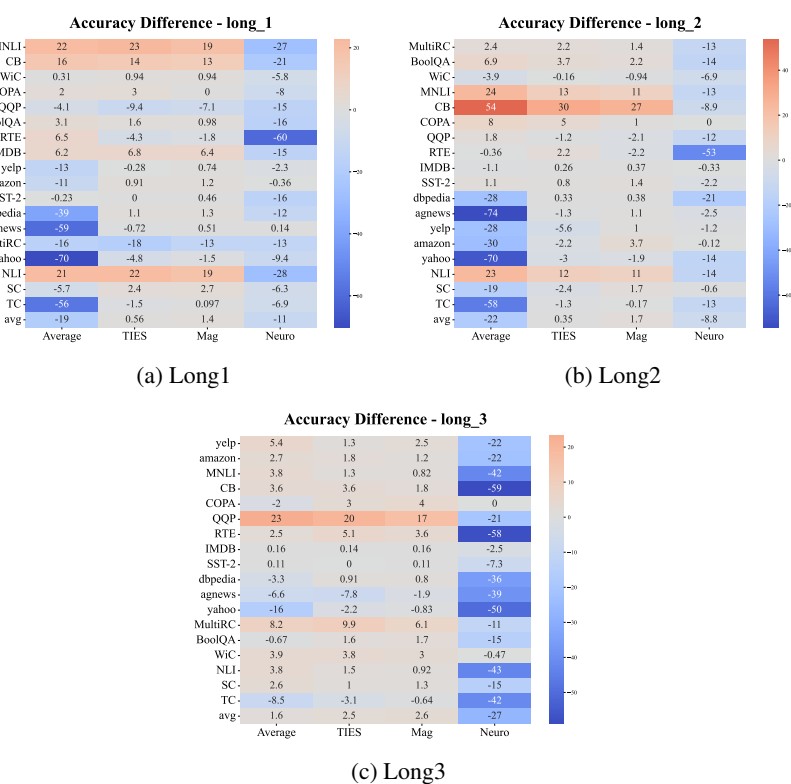

Figure 12: Heatmap showing the average accuracy change for each task before and after applying different merge method, evaluated after the final task in the long-order training sequence.

As illustrated in Figure 12, most tasks benefit from the magnitude-selected merging process. A few cases of accuracy drop suggest room for improving the merge strategy for better compatibility across tasks.

### D.3 REGULARIZATION COMPONENT ANALYSIS.

We further evaluate the impact of the proposed **Parameter Stability Loss** by ablating its two components in Eqn. (4): (i) *Magnitude-constraint*, which penalizes large-magnitude parameters to control forgetting, and (ii) *Sign-alignment*, which encourages alignment between the signs of the current task weights and the accumulated task vector; Table 12 shows that removing either component leads to notable performance drops, with the full regularization achieving the best balance between knowledge retention and new task acquisition. This validates our hypothesis that both sign alignment and magnitude control are crucial for stable continual adaptation.

Table 12: Ablation study on regularization strategies in continual LoRA training. We evaluate different components including sign-aware loss, magnitude restriction, and their combinations. Results are reported across three task orderings on two benchmarks.

| Method | Standard ($N = 4$) | | | | Long ($N = 15$) | | | |
|---|---|---|---|---|---|---|---|---|
| | Order1 | Order2 | Order3 | avg | Long1 | Long2 | Long3 | avg |
| IncLoRA | 68.6 | 59.7 | 75.0 | 67.8 | 60.3 | 60.5 | 53.2 | 58.0 |
| Magnitude-constraint | 72.2 | 73.2 | 77.6 | 74.3 | 63.8 | 64.7 | 68.5 | 65.7 |
| +Merging | 76.9 | 78.7 | 79.0 | 78.2 | 69.9 | 68.3 | 69.1 | 69.1 |
| Sign-alignment | 77.6 | 76.2 | 78.3 | 77.4 | 70.3 | 67.8 | 67.9 | 68.6 |
| +Merging | 79.1 | 78.4 | 79.0 | 78.8 | 70.0 | 67.2 | 72.1 | 69.8 |
| Both | 79.2 | 78.3 | 78.3 | 78.6 | 72.9 | 74.7 | 73.1 | 73.6 |
| +Merging | **80.0** | **79.1** | **79.6** | **79.6** | **74.2** | **76.5** | **75.7** | **75.5** |

### D.4 SENSITIVITY OF HYPER-PARAMETERS

We conducted a sensitivity analysis of the stability loss coefficient $\lambda$ on benchmarks with different task lengths ($N$) and task sequence orders. The results, shown in the following table, indicate that PS-LoRA is generally robust within a moderate range of $\lambda$ values. We noticed that values of $\lambda$ between 0.001 and 0.1 consistently yield state-of-the-art performance. Noticeable performance degradation occurs only at extreme values (e.g., 0.0001 or 10), suggesting that the proposed stability loss is not highly sensitive to $\lambda$. For the experiments reported in the main paper, $\lambda$ was chosen based on performance on the evaluation sets, with the best-performing value selected for each case.

Table 13: Results for different $\lambda$ values.

| $\lambda$ | Test ($N = 4$) | Test ($N = 15$) | Eval ($N = 4$) | Eval ($N = 15$) |
|---|---|---|---|---|
| 10 | 72.19 | 69.37 | 72.46 | 72.56 |
| 1 | 76.22 | 73.20 | 72.07 | 76.85 |
| **0.1** | 78.91 | **75.25** | 80.68 | **78.44** |
| 0.01 | 78.61 | 74.20 | 82.93 | 76.58 |
| **0.001** | **79.60** | 72.70 | **84.05** | 74.32 |
| 0.0001 | 78.51 | 66.83 | 82.01 | 71.83 |

### D.5 EFFECT OF STABILITY LOSS ON CONVERGENCE DYNAMICS

To further assess the effect of the stability loss on training dynamics, we compared PS-LoRA and the baseline IncLoRA on task order 1 of the Standard benchmark. For each new task, we recorded the training accuracy at several checkpoints (0%, 20%, ..., 100%) throughout optimization, as reported in Table 14.

The results indicate that the introduction of the stability loss does not lead to a slowdown in convergence. Both PS-LoRA and IncLoRA display nearly identical

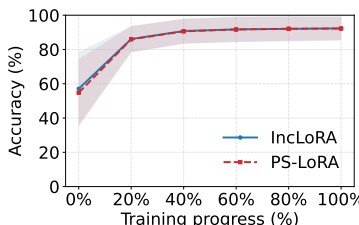

Figure 13: Convergence speed in training.

convergence behavior across tasks. For example, on $\mathcal{T}_4$,
both methods exceed 94% accuracy at 60% of training and converge to comparable final levels. In addition, PS-LoRA achieves consistently higher accuracy on all previously seen tasks, demonstrating improved retention without sacrificing optimization efficiency.

Table 14: Training accuracy of PS-LoRA and IncLoRA across checkpoints.

| Task | Method | 0% | 20% | 40% | 60% | 100% | All Seen |
|------|--------|------|------|------|------|------|----------|
| $\mathcal{T}_2$ | PS-LoRA | 52.94 | 73.54 | 78.56 | 79.33 | 80.55 | **79.34** |
| $\mathcal{T}_2$ | IncLoRA | 52.94 | 73.03 | 78.84 | 79.77 | 81.28 | 77.69 |
| $\mathcal{T}_4$ | PS-LoRA | 77.27 | 86.68 | 93.35 | 95.10 | 95.60 | **79.57** |
| $\mathcal{T}_4$ | IncLoRA | 75.70 | 87.33 | 93.03 | 94.34 | 95.23 | 62.62 |

## D.6 GPU MEMORY EFFICIENCY

We examined the GPU memory efficiency of PS-LoRA, which is a critical factor in continual learning with long task sequences. The evaluation was conducted from two perspectives: (i) parameter growth and (ii) runtime memory footprint. Across both, PS-LoRA demonstrates strong memory efficiency.

**Parameter growth.** LoRA adapters contribute only a small number of trainable parameters per task (Table 15). Even when adapters for all previous tasks are retained, the total parameter growth remains limited, especially in light of the substantial performance benefits. Compared with baselines, the parameter count of PS-LoRA is on pair with O-LoRA and IncLoRA, while significantly lower than MoE-style approaches such as AM-LoRA and MoCL, which require additional routing modules.

Table 15: Trainable LoRA parameters relative to full models.

| Backbone | Trainable | % of full model | Model Param |
|----------|-----------|-----------------|-------------|
| T5-large | 2.4M | **0.32%** | ~740M |
| LLaMA2-7B | 4.2M | **0.06%** | ~7B |

**Runtime memory.** We first compared PS-LoRA with IncLoRA and full fine-tuning in terms of latency and GPU memory usage. Batch sizes were set to 8 (training) and 128 (inference) on T5-large, and 4 (training) and 16 (inference) on LLaMA2-7B. Results are reported in Table 16.

Table 16: Training/inference latency and memory usage.

| Method | Training Latency | Inference Latency | Memory Usage |
|--------|------------------|-------------------|--------------|
| PS-LoRA (T5) | ~1.2s/it | ~4.3s/it | ~15GB |
| PS-LoRA Merged (T5) | – | **~2.1s/it** | ~15GB |
| IncLoRA (T5) | ~1.1s/it | ~4.3s/it | ~15GB |
| MoCL (T5) | ~1.3s/it | ~4.5s/it | ~30GB |
| Full-Finetune (T5) | ~1.4s/it | ~2.1s/it | ~20GB |
| PS-LoRA (LLaMA-7B) | ~1.7s/it | ~2.3s/it | ~30GB |
| PS-LoRA Merged (LLaMA-7B) | – | **~1.4s/it** | ~30GB |
| IncLoRA (LLaMA-7B) | ~1.5s/it | ~2.3s/it | ~30GB |
| Full-Finetune (LLaMA-7B) | ~12.0s/it | ~1.5s/it | ~45GB |

With regard to task length growth, we further recorded peak GPU memory usage (`max_allocated`) and static model footprint (`allocated`) across 15 tasks during both training and inference (Tables 17 and 18). The results show that PS-LoRA adds only a modest overhead, and memory consumption scales minimally with the number of tasks. This confirms that the stability-related parameters are not a dominant factor in runtime GPU usage.

## D.7 OTHER MERGING STRAGIES.

**How does the magnitude-based merging strategy in PS-LoRA compare to other merging strategies?** We evaluate different merging strategies, as shown in Table 19. Compared to alternatives such as simple averaging or Neuro Merging(Fang et al., 2025), our adopted merging strategy consistently

Table 17: GPU memory usage during LLaMA-7B training.

| | Task 1 | Task 3 | Task 6 | Task 9 | Task 12 | Task 15 | $\Delta_{\max}$ (1→15) |
|---|---|---|---|---|---|---|---|
| allocated (GB) | **12.64** | 12.68 | 12.67 | 12.72 | 12.72 | 12.74 | +110 MB (+0.9%) |
| max_allocated (GB) | **34.67** | 27.38 | 28.02 | 30.38 | 30.47 | 31.32 | **–3.35 GB (-9.7%)** |

Table 18: GPU memory usage during LLaMA-7B inference.

| | Task 1 | Task 3 | Task 6 | Task 9 | Task 12 | Task 15 | $\Delta_{\max}$ (1→15) |
|---|---|---|---|---|---|---|---|
| allocated (GB) | **12.63** | 12.65 | 12.67 | 12.69 | 12.72 | 12.74 | +110 MB (+0.9%) |
| max_allocated (GB) | **21.13** | 14.25 | 16.48 | 21.46 | 21.48 | 21.50 | **+0.37 GB (+1.8%)** |

achieves the best performance. This supports the intuition that parameters with larger magnitudes tend to be more important, which is consistent with the findings in Sec. 3.2. Therefore, the proposed PS-LoRA enhances performance by preserving the parameter distribution through the Parameter Stability loss, and by protecting high-magnitude parameters via the merging strategy.

Table 19: Ablation study on various post-training LoRA merging strategies based on T5-large. Results (%) are averaged over three random task orders on two continual learning benchmarks.

| **Merging Strategy** | **Standard** ($N = 4$) | | | | **Long** ($N = 15$) | | | |
|---|---|---|---|---|---|---|---|---|
| | Order1 | Order2 | Order3 | avg | Long1 | Long2 | Long3 | avg |
| None | 79.2 | 78.3 | 78.3 | $78.6_{\pm 0.5}$ | 72.9 | 74.7 | 73.1 | $73.6_{\pm 1.0}$ |
| Neuro(Fang et al., 2025) | 58.5 | 50.8 | 64.9 | $58.1_{\pm 7.1}$ | 61.9 | 66.5 | 55.4 | $61.3_{\pm 5.6}$ |
| Average | 77.0 | 78.1 | 77.3 | $77.0_{\pm 0.6}$ | 54.7 | 55.2 | 73.5 | $61.1_{\pm 10.7}$ |
| TIES(Yadav et al., 2023) | 78.6 | 78.3 | 79.6 | $78.8_{\pm 0.7}$ | 72.6 | 74.7 | 74.9 | $74.1_{\pm 1.3}$ |
| **Ours** | **80.0** | **79.1** | **79.6** | $\mathbf{79.6}_{\pm 0.5}$ | **74.2** | **76.5** | **75.7** | $\mathbf{75.5}_{\pm 1.2}$ |

## D.8 MERGING EFFICIENCY.

The merging operation in PS-LoRA is lightweight, based on an element-wise comparison between the accumulated LoRA weights and the current task's weights:

$$M(\Delta W_1, \Delta W_2)_{i,j} = \begin{cases} [\Delta W_2]_{i,j}, & \text{if } |\Delta W_2| \geq |\Delta W_1| \\ [\Delta W_1]_{i,j}, & \text{otherwise.} \end{cases} \quad (7)$$

This operation is highly parallelizable and scales linearly with the matrix size. For low-rank matrices $\mathbf{A}_i \in \mathbb{R}^{r \times d'}$ and $\mathbf{B}_i \in \mathbb{R}^{d \times r}$, where $r \ll d, d'$, the complexity is:

- **Time complexity:** $O(t \cdot r \cdot d \cdot d')$, with each step involving low-rank multiplications $O(r \cdot d \cdot d')$ and element-wise comparisons $O(d \cdot d')$. The small LoRA rank $r$ ensures efficiency even as the task count $t$ grows.
- **Space complexity:** only two intermediate tensors of size $d \times d'$ are needed, i.e., $O(d \cdot d')$ additional memory.

In practice, merging all 15 tasks requires only 0.28 seconds on T5-large and 1.47 seconds on LLaMA2-7B, confirming negligible cost even on large models.

**Runtime acceleration.** We benchmarked inference throughput (samples/sec) on the merged models. Merging yields a 40%–50% speedup, which is particularly beneficial for deployment:

**Reduced memory footprint.** Unlike MoE-style methods (e.g., MoCL, AM-LoRA) which cannot merge adapters, PS-LoRA significantly reduces memory usage. For instance, on T5-large, LoRA adapters introduce ∼2.4M parameters per task. Without merging, 15 tasks require 36M parameters (∼4.6% of the backbone size, 776M). With merging, the overhead is negligible.

Table 20: Inference throughput before and after merging.

| Model | w/o Merge | With Merge |
|---|---|---|
| T5-Large | 28.05 | **41.45** |
| LLaMA2-7B | 3.53 | **5.61** |

# E    STATISTICAL ANALYSIS

## E.1    FOR TABLES

We report standard deviation-based error bars for all results in Table 1, Table **??**, Table 5, Table 7, with Eqn.8

$$s = \sqrt{\frac{1}{n-1}\sum_{i=1}^{n}(x_i - \bar{x})^2} \tag{8}$$

This error bars in table are calculated across different orders, which reveal the stability for each method under different random task orders.

**Sources of Variability** The error bars capture variability due to three different orders.

**Method of Computation** Error bars were calculated as the standard deviation across 3 different orders.

## E.2    FOR GRAPHS

In Fig. 1, 7, we also report mean value and standard deviation-based error bars for all results by line plot and shadows.

**Sources of Variability** The error bars capture variability due to three different random seeds used for model initialization and data shuffling.

**Method of Computation** Error bars were calculated as the standard deviation across 3 runs with different seeds.

# F    LIMITATIONS AND FUTURE WORK

**Large Number Task and Efficiency** As the number of tasks increases, the memory overhead can become significant, despite the relatively small size of each low-rank matrix. This issue becomes particularly prominent in scenarios involving hundreds of tasks or when LoRA is injected into multiple layers of the model. A promising future direction is to investigate how to merge task-specific LoRA modules into the pretrained model incrementally during the continual learning process, or alternatively, maintain a single consolidated LoRA module that retains the knowledge acquired so far without catastrophic forgetting.

**Mechanism behind Sign Patterns** The underlying role of sign patterns in forgetting and learning dynamics remains insufficiently explored. Experimental results suggest that the sign components of LoRA parameters exhibit a certain degree of redundancy. Understanding how sign structures influence continual learning and parameter-efficient finetuning is crucial, as it may reveal fundamental mechanisms that drive knowledge retention and transfer in low-rank adaptation frameworks.

# G    THE USE OF LARGE LANGUAGE MODELS (LLMs)

In preparing this manuscript, we used a Large Language Model (LLM) as a writing assist tool. The LLM was employed solely for language polishing, including improving grammar, readability, and clarity of expression. It did not contribute to research ideation, experimental design, data analysis, or the generation of scientific content. The authors take full responsibility for the entirety of the manuscript.

