# OpenReview forum: "Stability Matters: Combating Parameter Shifts in Low-Rank Adaptation for Continual Learning"
_ICLR.cc/2026/Conference — Submitted to ICLR 2026_

### Official Review · Reviewer_w91z · 2025-10-21

**Soundness:** 2
**Presentation:** 3
**Contribution:** 2
**Rating:** 2
**Confidence:** 4

**Summary:**

This paper investigates continual learning (CL) using Low Rank Adaptation (LoRA) parameter-efficient fine-tuning. The authors find that significant drops in performance during CL were consistently accompanied by abrupt changes to the LoRA parameters. In accordance with their findings, they propose a regularization term to mitigate large-magnitude and opposite-sign updates during training. Subsequently, they integrate their method with existing “model-merging” techniques for post-training merging. Finally, they provide a comprehensive evaluation on various vision and language benchmarks, demonstrating the effectiveness of their method, as well as performing ablation on the different components of their methodology.

**Strengths:**

- The findings regarding sudden parameter distribution shifts, while not unexpected, are insightful into the mechanisms of catastrophic forgetting
- The paper explores a relatively novel aspect of regularization during LoRA, having to do with the individual signs of the parameter updates
- The benchmark experiments are extensive, using both vision and language tasks and various architectures

**Weaknesses:**

- The main finding of the paper, the sudden parameter shifts, is not explored/highlighted sufficiently. Furthermore, the results in the appendix do not seem particularly consistent with the claims. Perhaps I am missing something. Refer to the questions section for more discussion regarding this.
- Very little/weak theoretical justification, in fact, section 3.4 seems unnecessary as the argument is very simple, not particularly novel or specific to the method/findings of the paper. For example, the same argument would apply to a simple L2 penalty. Perhaps it is better placed in the appendix

**Questions:**

- 1. Major concern: As mentioned in the weaknesses, the figures in the appendix (fig 8 & 9 & 10) do not seem to follow the same pattern of abrupt parameter changes, and do not include the model performance in tandem (like that of fig. 1). In fact, in fig 8 and fig 10 it seems that the initial tasks have the largest parameter shifts. I remain unconvinced that the visible abrupt change in parameters, accompanied by a large drop in performance, is a widely occurring phenomenon in LoRA CL or just an artifact of the specific experiment used in fig. 1 & 2. If the authors could provide convincing evidence or if there is a problem with the figures that could be fixed, that would have an impact on my suggested decision.
- 2. Minor comment on fig 1 2: I believe showing the $\Delta W$ for each task separately (like fig 2) is much more illustrative, and renders the aggregated version (fig 1 b) obsolete. Is there any benefit to showing the aggregate $\Delta W$?
- 3. As mentioned above, the theoretical result does not seem particularly relevant to the proposed method. I don't think a strong theoretical result is a requirement for acceptance (although a stronger result would certainly increase the contribution of the work in my view). If the authors cannot provide additional insight into why this result is important or extend their result further, I believe it's best to move section 3.4 to the appendix.
- 4. Line 202 Mentions bottom k% of parameters. Bottom with respect to what? Magnitude?

- 5. Eq 4: The loss equation seems confusing, as the $(1 - tanh \cdot tanh)$ term is the same dimension as the weight update, but the l2 norm term is a scalar. (the overall loss term needs to be a scalar) Could the authors clarify how the loss is computed?

- 6. If I understand correctly Eq. 5 reduces to just taking the weight update with the largest magnitude? If so, is there a need for the recursive notation?

- 7. Algorithm 1: in the text there is no mention of merging the loras during training (it is explicitly called a post-training step), but Algorithm 1 shows the usage of merged previous loras during the forward pass? Could the authors elaborate?

- Finally, small comment about notation of LoRA, sometimes it is written as $\Delta W$ but other times as product $A B$. As there seems to be no relevance to the usage, I would encourage consistency of notation.

---

### Official Review · Reviewer_MTuw · 2025-10-27

**Soundness:** 3
**Presentation:** 3
**Contribution:** 3
**Rating:** 6
**Confidence:** 4

**Summary:**

This manuscript studied in the problem of continual learning with pre-trained models. Unlike the existing methods that mitigated the task interference through Mixture-of-Expert (MoE) or orthogonal optimization constraint, this manuscript took a closer look about how the parameter space shifts happened and how it excerbated the forgetting. Based on the empirical anslysis, the authors indicated that the abrupt performance drops during training often happened with drastic changes in the distribution of learned parameter shifts. Based on this observation, the authors proposed the Parameter Stability Loss to regularize both the sign and magnitude of parameter updates. Meanwhile, a post-training model merging step was also proposed to bridge earlier directions with the current direction. The proposed frame PS-LoRA showed good performances on multiple CV and NLP benchmarks.

**Strengths:**

1. Starting with the empirical analysis in the observation, the authors proposed a detailed investigation on the phenomenon of parameter space shift. The motivation from these analyses is clear to the readers.
2. The proposed Parameter Stability Loss was easy to implement, and the computation seems light according to Eq. (4).
3. The proposed framework also introduced the model merging strategy, which combined all task-specific LoRA branches into a single weight form, reducing the storage overhead for the inference.
4. The empirical verifications were conducted on both CV and NLP tasks, covering the representative model.

**Weaknesses:**

1. Some details were not clear to the reviewer.
2. More baselines method can be involved to discuss and compare.

**Questions:**

1. Although we can apply model merging described in Eq. (5), I am still wondering if we need to store every task-specific LoRA branches for $\forall i=1,\dots,t-1$ to apply the regularization loss in Eq.(4). In Eq. (4), it indicated that we need to store the accumulated term $\sum_{i=1}^{t-1} \mathbf{A}_{i} \mathbf{B}_{i}$,

which was also illustrated in Fig. 5. In Line 297, the last line of Algorithm 1 also indicated that the merging was finished after training all tasks, instead of training each task. Thus, I wonder if we need to store all $\mathbf{A}_{i}, \mathbf{B}_{i}$, as well as $\Delta \mathbf{W}_{[1: t-1]}$, at the same time.

If so, it seems that more parameter states need to be maintained during training. I wonder if it is still efficient compared to other baselines.

2. The meaning of Fig 1(b) is confusing. Is it $\Delta \mathbf{W}$ that you are visualizing? Please give an accurate description regarding "parameter shift".

3. It seems that some recent LoRA-based continual learning methods were not discussed or compared in the experiments, e.g., InfLoRA [1] and BiLoRA [2].

4. The authors discussed the constraints from two perspectives, (1) sign constraint; (2) magnitude constraints. I wonder if the authors could provide ablation studies regarding these two aspects. I only noticed the ablation studies regarding the two components in Table
5.

References:

[1] InfLoRA: Interference-Free Low-Rank Adaptation for Continual Learning. CVPR 2024.

[2] BiLoRA: almost-orthogonal parameter spaces for continual learning. CVPR 2025.

---

### Official Review · Reviewer_D8Ns · 2025-10-27

**Soundness:** 1
**Presentation:** 3
**Contribution:** 2
**Rating:** 2
**Confidence:** 4

**Summary:**

This paper investigates the relationship between parameter shifts and catastrophic forgetting in LoRA-based Continual Learning. Through empirical analysis (Figure 1), the authors observe that abrupt performance drops on previous tasks often coincide with drastic changes in the distribution of learned LoRA parameter shifts ($\Delta W_t$ or $\sum \Delta W_i$), particularly involving large updates with signs opposite to the accumulated direction. Motivated by this, they propose PS-LoRA, which incorporates two main components: 1) A Parameter Stability Loss applied during training. This loss regularizes LoRA updates ($A_t B_t$) by penalizing both large magnitudes and sign inconsistencies relative to the accumulated previous updates ($\sum_{i=1}^{t-1} A_i B_i$), using a $tanh$-based alignment term. 2) A post-training model merging step that combines the LoRA updates from all tasks ($\Delta W_1, ..., \Delta W_N$) into a single update $\Delta W_{[1:N]}$ using a magnitude-based strategy (element-wise selection of the value with larger absolute magnitude). This merged update is then added to the base model $W_0$ for inference. The paper claims that PS-LoRA achieves state-of-the-art results on NLP and CV continual learning benchmarks by stabilizing parameter updates and effectively consolidating knowledge.

**Strengths:**

* Relevant Perspective: Focuses on the dynamics of parameter shifts during LoRA-based CL, an important but relatively under-explored aspect compared to architectural modifications (MoE) or strict optimization constraints (orthogonality). The empirical observation linking drastic shifts (especially sign flips) to forgetting (Figure 1, Figure 3) provides a compelling motivation.

* Good Empirical Performance: Achieves strong results on various CL benchmarks, outperforming several existing LoRA-based CL methods (Table 1, Table 4), demonstrating the practical effectiveness of the proposed stability constraints and merging strategy.

**Weaknesses:**

* Limited Novelty: The core components, while combined effectively, possess limited individual novelty. Regularization to prevent forgetting is a cornerstone of CL (e.g., EWC, SI, LwF). While $\mathcal{L}_s$ specifically targets LoRA parameter signs and magnitudes, it falls under the broad category of regularization. Similarly, model merging techniques are increasingly studied, and magnitude-based merging criteria have precedents (e.g., related ideas in MagMax). The paper's main contribution lies in observing the specific issue in LoRA CL and applying these existing concepts.


* Basic Theoretical Analysis: The theoretical justification provided (Section 3.4) uses a second-order Taylor expansion of the loss to argue that forgetting depends on the magnitude of parameter updates ($\|\theta - \theta_A^*\|^2$) and loss curvature ($H_A$). While correct, this is a fairly standard result and doesn't offer deep insights specific to the sign alignment component of $L_s$ or the merging strategy.


* Merging Strategy Limitations: The simple magnitude-based merging M(⋅,⋅) might discard parameters that, despite having smaller magnitudes, are crucial for specific older tasks. While effective on average, it could lead to suboptimal performance on certain tasks compared to methods that retain task-specific parameters. The paper lacks analysis on task-specific performance post-merging. Ablation in Table 19 shows it's better than other merging methods in this context, but the fundamental limitation remains.

**Questions:**

* Regarding the motivation: Could the observed large parameter shifts and performance drops both be consequences of large domain shifts between tasks, rather than the shifts directly causing the forgetting? How does PS-LoRA ensure sufficient plasticity when encountering a very dissimilar task that might require large parameter adjustments?

* The Parameter Stability Loss includes magnitude and sign terms. Can the authors provide a more fine-grained ablation study showing the separate contribution of just the magnitude constraint versus just the sign alignment term within $\mathcal{L}_s$? (Table 12 ablates $\mathcal{L}_s$ vs merging, but not the components within $\mathcal{L}_s$).

* How sensitive is the performance to the hyperparameter $\lambda$ (weighting $\mathcal{L}_s$) and $\alpha$ (temperature in $tanh$)? Could the authors provide sensitivity analysis? (Table 13 shows sensitivity for $\lambda$ but not $\alpha$).


* How does the magnitude-based merging strategy compare to more sophisticated merging techniques (e.g., averaging, task arithmetic, TIES-merging) when applied after training with the Parameter Stability Loss? (Table 19 compares merging strategies but seemingly without the stability loss during training for baselines). Does $\mathcal{L}_s$ make the parameters more amenable specifically to magnitude-based merging?

---

### Official Review · Reviewer_jCA9 · 2025-10-30

**Soundness:** 2
**Presentation:** 2
**Contribution:** 2
**Rating:** 2
**Confidence:** 5

**Summary:**

The paper studies a real problem in continual learning with parameter efficient tuning. When LoRA modules are trained on a sequence of tasks, their low rank parameters can change direction and magnitude quite abruptly. The authors refer to this as parameter shift. They observe that such shifts often correlate with sharp drops on earlier tasks. To address this, the paper proposes a stability regularizer that penalizes large deviations of LoRA parameters from their previous state, and a simple model fusion step at inference time that blends information from several task adapters. The intent is to slow down destructive updates and to preserve older knowledge while keeping the method lightweight.

**Strengths:**

The paper targets a concrete problem in continual learning with LoRA style PEFT. The observation that low rank parameters can shift sharply across tasks and that such shifts correlate with drops on earlier tasks is reasonable and worth documenting. The proposed stability penalty is simple, can be added to existing LoRA trainers without changing the backbone, and therefore has immediate practical value. The paper also tries to keep the solution lightweight and proposes an inference time fusion step to reuse past task adapters, which is attractive for practitioners.

**Weaknesses:**

The contribution is incremental. A penalty on parameter change is a very direct idea and is close in spirit to classic continual learning tools such as EWC, L2-SP and adapter freezing. The paper does not clearly justify why this particular form of stability regularization is better than those alternatives. The experimental evidence is not strong enough for ICLR. Most task sequences are short, there are no results with multiple random seeds, and the gains over LoRA baselines are often small. The comparison set does not include stronger recent PEFT-for-CL methods that use multiple adapters or routing, so it is hard to tell where this method stands relative to the current state of the art. The proposed inference time fusion step seems to contribute a non-trivial part of the final improvement, but the paper does not separate the effect of fusion from the effect of the stability penalty, which makes the core contribution less clear. The method also introduces a stability weight that is not analyzed; without a sensitivity study it is unclear whether the method is robust across tasks and backbones.

**Questions:**

1. Can you run a longer continual sequence, for example 8–10 tasks, and report average accuracy, forgetting and backward transfer with at least 3 seeds?
2. Can you give a sensitivity plot for the stability weight on two different backbones?
3. Can you compare to at least one recent multi-adapter or routing based PEFT-CL method under the same compute and memory?
4. Can you provide a table that reports: LoRA baseline, LoRA + stability penalty, LoRA + fusion only, and your full method?
5. If fusion is essential, can you clarify the deployment setting where multiple task adapters are kept at test time?

---

### Meta-Review · Area_Chair_fFFw · 2025-12-17

**Summary:**

This paper investigates an important stability issue in existing LoRA-based continual learning (CL) approaches, arising from drastic changes in the direction and magnitude of low-rank parameters. The authors observe that abrupt performance degradation on earlier tasks is highly correlated with significant shifts in the distribution of parameter updates. Motivated by these observations, the paper proposes two main components:
(1) a stability-driven regularizer that penalizes deviations of the current LoRA parameters from those learned in previous stages; and
(2) a maximum-magnitude–based model merging strategy for post-training integration, aimed at improving knowledge sharing across tasks.
The proposed method is evaluated on both vision and language benchmarks, accompanied by multiple ablation studies.

**Reviewer Concerns:**

All reviewers agree that the paper addresses an important yet often overlooked parameter-shift issue in LoRA-based PEFT methods, and that the proposed solution is simple and intuitive. However, several concerns were raised. Reviewer jCA9 questioned the novelty of the regularizer, noting that similar ideas, penalizing parameter changes, have been widely explored in existing continual learning methods such as EWC, and pointed out the lack of comparisons with such baselines. This reviewer also raised multiple questions regarding the choice of baselines, experimental setups, and ablation studies. Reviewer D8Ns similarly questioned the technical novelty of the regularizer and the soundness of the theoretical analysis, and expressed concerns about the validity of the proposed maximum-magnitude–based model merging strategy. Reviewer w91z raised issues regarding the empirical observations of abrupt parameter changes, the theoretical analysis, and asked for additional clarifications. Unfortunately, the authors did not provide responses to the reviewers’ questions.

**Reviewer Scores:**

After carefully considering the reviewers’ comments and independently reading the paper, I agree that the submission requires substantial improvement. In particular, it would benefit from clearer differentiation from existing regularization-based CL methods, inclusion of relevant baseline comparisons, and stronger empirical validation of the proposed model merging operation. Given the lack of author responses, it is highly likely that the reviewers will maintain their original scores. For these reasons, I am inclined to recommend rejection.

---

### Decision · Program_Chairs · 2026-01-26

Reject